# Memory for nonadjacent dependencies in the first year of life and its relation to sleep

Manuela Friedrich ®[1,2] ✉, Matthias Mölle ®[3], Jan Born ®[4] & Angela D. Friederici ®[2]

Grammar learning requires memory for dependencies between nonadjacent elements in speech. Immediate learning of nonadjacent dependencies has been observed in very young infants, but their memory of such dependencies has remained unexplored. Here we used event-related potentials to investigate whether 6- to 8-month-olds retain nonadjacent dependencies and if sleep after learning affects this memory. Infants were familiarised with two rule-based morphosyntactic dependencies, presented in sentences of an unknown language. Brain responses after a retention period reveal memory of the nonadjacent dependencies, independent of whether infants napped or stayed awake. Napping, however, altered a specific processing stage, suggesting that memory evolves during sleep. Infants with high left frontal spindle activity show an additional brain response indicating memory of individual speech phrases. Results imply that infants as young as 6 months are equipped with memory mechanisms relevant to grammar learning. They also suggest that during sleep, consolidation of highly specific information can co-occur with changes in the nature of generalised memory.

The ability to detect relationships between spatially separated or temporally distant elements of one's environment is a core component of the mental structure-building process in humans. Memory of these nonadjacent relationships is crucial for causal reasoning, language, and other kinds of hierarchical mental processing. Despite its significance for higher mental processing, nonadjacent dependency (NAD) learning is not a unique human ability, also being reported in non-human species such as apes, monkeys, and songbirds[1–4]. It may have evolutionary roots in processes of sensory-motor integration and motor control that enable vocal imitation and action planning[5,6].

In humans, memory for temporally organised, rule-based patterns in the auditory environment is a prerequisite of grammar learning. Brain responses in infants have shown that even 4-month-olds are able to encode NADs in spoken sentences of an unknown natural language[7]. At 7 months, infants show first behavioural evidence of tracking NADs in artificial grammars[8,9]. However, it has not yet been studied

whether infants in their first year of life retain nonadjacent regularities in memory.

Sleep supports memory in both the mature and the immature brain[10–22]. Studies of early lexical-semantic learning have shown that sleep following the encoding of object-word pairs is crucial for infants to form memory of the dependencies between the objects and words[11,13,15,16]. Sleep spindles, short periods of 10–15 Hz oscillatory activity during NonREM sleep (REM = Rapid Eye Movement), indicate brain states of increased synaptic plasticity that are involved in the sleep-dependent consolidation of memories[10,17–21]. In infants aged 6 to 16 months, sleep spindles in central and parietal brain regions support lexical-semantic generalisation from a set of object-word pairs in which similar objects are paired with the same word[11,15,16]. Moreover, sleep spindles in frontal brain regions in 14- to 17-month-olds are related to the consolidation of detailed memory for the episodic occurrence of a specific object with a certain word[22].

[1]Department of Psychology, Humboldt-Universität zu Berlin, Rudower Chaussee 18, D – 12489 Berlin, Germany. [2]Department of Neuropsychology, Max Planck Institute for Human Cognitive and Brain Sciences, Stephanstraße 1a, D – 04103 Leipzig, Germany. [3]Center of Brain, Behavior and Metabolism (CBBM), University of Lübeck, Ratzeburger Allee 160, D – 23562 Lübeck, Germany. [4]Institute of Medical Psychology and Behavioral Neurobiology and Center for Integrative Neuroscience, University of Tübingen, Otfried-Müller-Str. 25, D – 72076 Tübingen, Germany. ✉e-mail: friedri@cbs.mpg.de

Encoding phase: La sorella **sta** cant–**ando**. Il fratello **può** cant–**are**.
 A X B C X D

Memory test phase:

(1) **Regular** structure (A–B, C–D), **old** verb stems (X) : La sorella **sta** cant–**ando**. Il fratello **può** cant–**are**.
 A X B C X D

(2) **Irregular** structure (A–D, C–B), **old** verb stems (X): Il fratello **può** cant–ando. La sorella **sta** cant–are.
 C X B A X D

(3) **Regular** structure (A–B, C–D), **new** verb stems (Y): La sorella **sta** parl–**ando**. Il fratello **può** parl–**are**.
 A Y B C Y D

(4) **Irregular** structure (A–D, C–B), **new** verb stems (Y): Il fratello **può** parl–**ando**. La sorella **sta** parl–**are**.
 C Y B A Y D

**Fig. 1 | Experimental design.** In the encoding phase, verb phrases of the form AXB and CXD with fixed nonadjacent structure **A**–**B** and **C**–**D** and varying intermediate verb stems X were presented in short sentences. In the memory test phase, infants were exposed to: (1) regular sentences with the same nonadjacent structure and the same old verb stems as in the encoding phase (AXB, CXD), (2) sentences with irregular structure, which violated the familiarised regularities, but contained the old verb stems of the encoding phase (AXD, CXB), (3) sentences with regular structure, but new verb stems (AYB, CYD), and (4) sentences with irregular structure and new verb stems (AYD, CYB). The probability of occurrence of elements (**A**, **B**, **C**, and **D**) was equal in all conditions. In addition, all transitional probabilities of adjacent elements were kept constant.

In adults, lexical-semantic memory and detailed episodic memory are both part of the consciously available, hippocampus-dependent declarative memory, whereas native-language syntactic abilities are primarily based on non-declarative memory[23,24]. To date there is no evidence for a similar distinction between these kinds of language-related memories during the earliest, preverbal stages of development. It is not yet specified whether the earliest memory for regularities in object-word pairings and the earliest memory for syntactic regularities differ qualitatively or whether they share a common neural basis and use the same mechanisms of memory formation and generalisation. In particular, initial behavioural evidence suggests that sleep modulates NAD memory in 15-month-olds[25,26], but it is not known if sleep affects the memory for syntactic NADs in much younger infants and whether or not its consolidation differs from that of lexical-semantic memory.

In the present study, we utilised event-related potentials (ERPs) to assess the learning and memory of NADs in 6- to 8-month-olds, and analysed nap sleep patterns during the retention period to uncover the relationship between sleep and early forms of morphosyntactic memory. In an encoding phase, infants were familiarised with two NADs of the form AXB and CXD, presented in short sentences of an unknown language. A and C were auxiliary and modal verbs respectively, X was a varying verb stem, and B and D verb suffixes that were dependent on the occurrence of either A or C[7,27]. In the retention period following familiarisation, infants of one group napped while infants of a second group stayed awake. In the memory test phase after the retention period, infants heard regular sentences with the familiarised NADs (A–B and C–D) as well as irregular sentences that violated the familiarised structure (A–D and C–B). Half of the sentences contained an old intermediate verb stem of the encoding phase (X) to test for retention, the other half a new verb stem (Y) to test for generalisation (Fig. 1).

Our results show that 6- to 8-month-old infants retain their familiarity with NADs in memory and generalise the nonadjacent regularities to novel verb stems. Unlike lexical-semantic memory examined in previous infant studies[11,15], sleep after encoding is not crucial for this earliest memory of NADs observed so far. However, napping modifies subsequent NAD processing, suggesting that memory of the regularities further evolves during sleep. In addition, we find that spindle activity in the left frontal cortex is associated with a memory that is specific to the old speech phrases presented in the encoding phase. We conclude that both the mechanisms of forming and evolving generalised memory for nonadjacent regularities and a sleep-dependent mechanism of consolidating highly

specific memory are effectively established within the first half year of life.

## Results

In a first step, we analysed the data of the encoding phase in order to find out how the infants' brain responses to morphosyntactic NADs change with increasing familiarity. In particular, we compared the ERPs of the verbs' suffixes in the first half of the encoding phase with those of the second half. In a next step, we analysed the data of the memory test phase to uncover (1) how the ERP responses to the same suffixes differ between sentences with regular (familiarised) and irregular structure, (2) if this difference depends on whether or not a verb stem was already presented in the encoding phase, and (3) to what extent a nap in the retention period affects the memory effects in the infant ERP. Note that the polarity (i.e., positivity or negativity) of an ERP difference is relative and depends on the polarity of the initial deflection, the direction of change, and the direction of subtraction. Here we present the learning-induced changes of the initial deflections and further describe ERP differences in such a way that they can be linked to language-specific and memory-related ERP components known from previous studies in infants and adults.

### Same familiarity effects in the wake and nap groups

The formation of immediate memory during the encoding session was reflected in three familiarity effects, which occurred at different latencies in the ERP (Fig. 2a). First, the initially pronounced, frontal to central early-latency positivity decreased with increasing familiarity ($F_{1,84} = 12.324$, $P = 0.001$, $\eta_p^2 = 0.128$, Familiarity × Region $F_{2,168} = 6.976$, $P = 0.001$, $\eta_p^2 = 0.077$, repeated measures ANOVA; frontal $t_{84} = -4.072$, $P = 0.0002$, $d = -0.442$, 95% CI = [−2.993, −1.029], central $t_{84} = -3.565$, $P = 0.001$, $d = -0.387$, 95% CI = [−2.769, −0.786], one-sample $t$-tests). This effect is well-known as a familiarity-based increase in the N200–500 component of the infant ERP[28–33], whereby the shortened latency here results from calculating ERPs relative to suffix-onset. It indicates changes in word processing due to the formation and successive strengthening of the acoustic-phonological representations of the suffixes. Second, a mid-latency positivity with a mid-central maximum decreased from the first to the second half of the encoding session ($F_{1,84} = 4.622$, $P = 0.034$, $\eta_p^2 = 0.052$, mid-central $t_{84} = -2.651$, $P = 0.010$, $d = -0.288$, 95% CI = [−2.791, −0.398]). This decreased positivity is similar to the effect observed for the immediate learning of NADs in 4-month-olds[7], which in the 6- to 8-month-olds here, occurs earlier than in the younger infants. Thirdly, a late frontal negativity decreased with increasing familiarity ($F_{1,84} = 4.344$, $P = 0.040$,

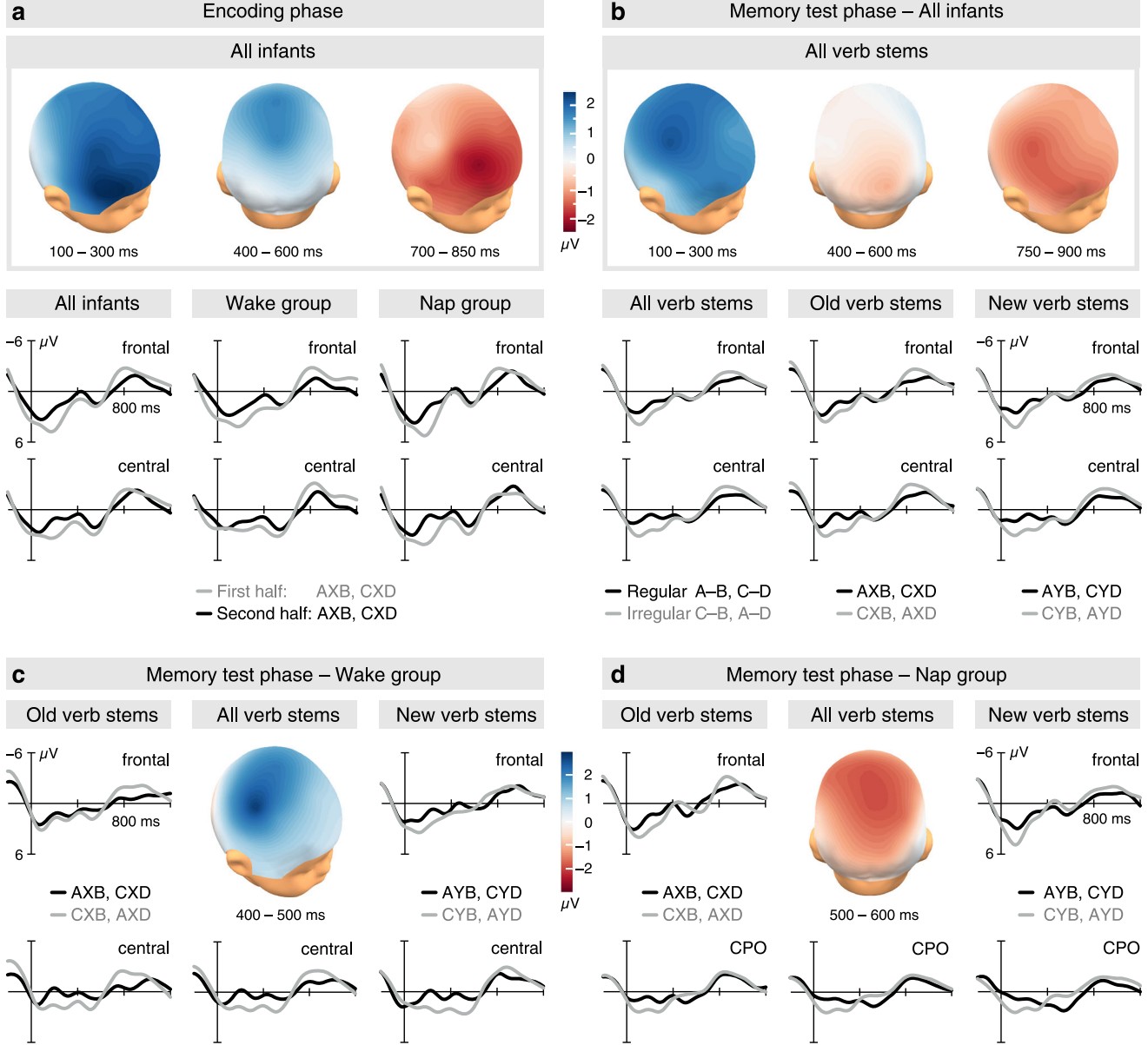

**Fig. 2 | Familiarity effects and memory effects.** ERPs and the spatial distribution of the effects of the encoding and memory test phases time-locked to the onset of the suffixes (**b** and **d**). Negativity is plotted upward. Source data are provided as a Source Data file. For responses at individual electrode positions, see Supplementary Figs. 1–3. **a** The three familiarity effects of the encoding phase in the overall group ($N = 85$ infants), and the ERPs over the frontal and central regions for the first half (grey lines) and the second half (black lines) of the encoding phase in the overall group, the wake group ($N = 37$ infants), and the nap group ($N = 48$ infants). Voltage maps represent ERP differences between the responses in the first and the second half of the encoding session. **b** The memory effects at early and late

latencies, no memory effect at middle latency in the overall group ($N = 85$ infants), and the ERPs over frontal and central regions for regular (black lines) and irregular (grey lines) sentences averaged across all phrases, phrases with old verb stems, and phrases with new verb stems. Voltage maps represent ERP differences between irregular and regular sentences. **c** The mid-latency memory effect in the wake group ($N = 37$ infants) and ERPs over frontal and central regions for old and new verb stems. Voltage maps as in 2b. **d** The mid-latency memory effect in the nap group ($N = 48$ infants) and ERPs over frontal and CPO regions for old and new verb stems. The CPO region included the central and parietal-occipital regions. Voltage maps as in 2b.

$\eta_\mathrm{p}^2 = 0.049$, Familiarity × Region $F_{2,168} = 3.974$, $P = 0.021$, $\eta_\mathrm{p}^2 = 0.045$; frontal $t_{84} = 3.080$, $P = 0.003$, $d = 0.334$, 95% CI = [0.541, 2.512]). This late effect most likely represents the infant-specific Nc component, which has been observed in a variety of infant ERP studies and is thought to reflect memory-related changes in the infants' attention[34,35]. None of the effects of the encoding session were related to the infants' age (early frontal: $r = -0.107$, $P = 0.331$, 95% CI = [−0.313, 0.109], mid-latency mid-central: $r = 0.079$, $P = 0.472$, 95% CI = [−0.136, 2.87], late frontal: $r = 0.110$, $P = 0.317$, 95% CI = [−0.106, 0.316]). Moreover, the effects did not differ in amplitude between the wake and nap groups (early: $t_{83} = 0.663$, $P = 0.509$, 95% CI = [−1.326, 2.650], mid-latency:

$t_{83} = -0.111$, $P = 0.912$, 95% CI = [−2.563, 2.292], late: $t_{83} = -0.030$, $P = 0.976$, 95% CI = [−2.151, 2.089], $t$-test for independent samples; Fig. 2a).

**Retention and generalization in the wake and nap groups**
In the memory test session after the retention period, two of the ERP effects that indicated increased familiarity in the encoding session were observed as memory effects. In particular, the suffixes in regular sentences were processed as if they were more familiar than exactly the same suffixes in irregular sentences (Fig. 2b). The early-latency memory effect (Regularity $F_{1,83} = 14.172$, $P = 0.0003$, $\eta_\mathrm{p}^2 = 0.146$,

repeated measures ANOVA) was statistically verifiable in separate analyses of each group (wake: $t_{36} = 2.602$, $P = 0.013$, d = 0.428, 95% CI = [0.301, 2.431], nap: $t_{47} = 2.844$, $P = 0.007$, $d = 0.410$, 95% CI = [0.394, 2.300], one-sample $t$-test) and did not differ in amplitude between them ($t_{83} = 0.027$, $P = 0.979$, 95% CI = [−1.392, 1.430], $t$-test for independent samples). Also, the effect was present for both old and new verb stems (old: $t_{84} = 2.765$, $P = 0.007$, $d = 0.300$, 95% CI = [0.385, 2.357], new: $t_{84} = 2.710$, $P = 0.008$, $d = 0.294$, 95% CI = [0.356, 2.323], one-sample $t$-test), with no difference between these separate effects ($t_{84} = 0.044$, $P = 0.965$, 95% CI = [−1.363, 1.425], $t$-test for dependent samples). Similarly, the late memory effect (Regularity $F_{1,83} = 7.304$, $P = 0.008$, $\eta_{p}^{2} = 0.081$) differed neither between wake and nap groups ($t_{83} = −0.754$, $P = 0.453$, 95% CI = [−1.694, 0.763], $t$-test for independent samples) nor between the suffixes following old and new verb stems ($t_{84} = −0.335$, $P = 0.738$, 95% CI = [−1.873, 1.333], $t$-test for dependent samples). The memory effects at early and late latencies were correlated ($r = −0.461$, $P = 0.00001$, 95% CI = [−0.614, −0.275]) and the strength of the correlations did not differ between groups ($z = −0.576$, $P = 0.565$; wake: $r = −0.519$, $P = 0.001$, 95% CI = [−0.722, −0.234], nap: $r = −0.417$, $P = 0.003$, 95% CI = [−0.627, −0.151]). None of the effects were related to age (early: $r = −0.018$, $P = 0.867$, 95% CI = [−0.231, 0.195], late: $r = 0.151$, $P = 0.167$, 95% CI = [−0.064, 0.353]). Furthermore, we found no significant correlations between the memory effects and individual sleep times (early: $r = −0.088$, $P = 0.551$, 95% CI = [−0.363, 0.201], late: $r = 0.208$, $P = 0.157$, 95% CI = [−0.081, 0.464]).

These processing differences for the same suffixes in different nonadjacent contexts indicate the retention of the familiarised NADs. Their independence from the intervening verb stem implies generalised memory of the regularities. The presence of same memory effects in both groups strongly suggests that sleep after encoding is not necessary for the memory of morphosyntactic NADs in 6- to 8-month-old infants.

## Memory evolvement in the nap group

The analyses of the mid-latency ERP segment of the memory test phase revealed processing differences between the wake and nap groups (400−600 ms: Regularity × Group $F_{1,83} = 9.309$, $P = 0.003$, $\eta_{p}^{2} = 0.101$, Regularity × Region × Group $F_{2,166} = 3.476$, $P = 0.043$, $\eta_{p}^{2} = 0.040$, repeated measures ANOVA). The wake group showed a less positive ERP response over the central region for suffixes in regular compared to irregular sentences ($t_{36} = 2.491$, $P = 0.017$, $d = 0.410$, 95% CI = [0.282, 2.753], one-sample $t$-test). The effect was similar to that seen with increasing familiarity in the encoding session and had a maximum over the right central region within 400 and 500 ms post suffix onset ($t_{36} = 2.913$, $P = 0.006$, $d = 0.479$, 95% CI = [0.678, 3.789]; Fig. 2c). This maximum did not differ between old and new verb stems ($t_{36} = −0.643$, $P = 0.525$, 95% CI = [−4.133, 2.144], $t$-test for dependent samples). It was neither correlated with the infants' age ($r = −0.081$, $P = 0.634$, 95% CI = [−0.395, 0.250]), nor with the duration of the retention period ($r = 0.005$, $P = 0.975$, 95% CI = [−0.329, 0.319]). The right-central mid-latency memory effect was however positively related to the mid-latency familiarity effect observed during encoding ($r = 0.410$, $P = 0.012$, 95% CI = [0.099, 0.648]). The occurrence of a total of three similar ERP effects in the encoding and memory test phases in the wake group suggests that infants in this group retained their immediately formed representations of the NADs in memory and, moreover, that the NAD phrases underwent similar processing stages as during encoding when infants were re-exposed to them in the memory test.

In contrast, the nap group displayed a distinct, polarity-inversed mid-latency memory effect ($t_{47} = −2.550$, $P = 0.014$, $d = −0.368$, 95% CI = [−1.621, −0.191], one-sample $t$-test), with a positive ERP wave for suffixes in regular sentences and a less positive response for those in irregular sentences. The relative negativity for irregular sentences was strongest between 500 and 600 ms over the central ($t_{47} = −3.031$, $P = 0.004$, $d = −0.438$, 95% CI = [−2.439, −0.493]) and parietal-occipital

($t_{47} = −3.289$, $P = 0.002$, $d = −0.475$, 95% CI = [−1.866, −0.450]) regions. The mean effect over these regions (Fig. 2d) was neither related to the infants' age ($r = −0.053$, $P = 0.720$, 95% CI = [−0.332, 0.235]) nor to the early ($r = −0.227$, $P = 0.120$, 95% CI = [−0.481, 0.061]) or late ($r = −0.085$, $P = 0.565$, 95% CI = [−0.361, 0.204]) memory effects of this group. It was also not correlated with any of the effects of the encoding phase (early: $r = −0.121$, $P = 0.414$, 95% CI = [−0.391, 0.169], mid-latency: $r = −0.010$, $P = 0.946$, 95% CI = [−0.275, 0.293], late: $r = −0.195$, $P = 0.185$, 95% CI = [−0.454, 0.095]). Compared to the processing of NADs during encoding, this mid-latency memory effect of the nap group represents a qualitative shift in a particular stage of NAD processing, implying a modification of the underlying memory representations during the retention period. In the wake group, individual retention time was not correlated with the respective ERP difference ($r = 0.103$, $P = 0.542$, 95% CI = [−0.228, 0.414]), rendering unlikely that the overall longer retention time in the nap group was responsible for this effect. The appearance of a new processing stage in the nap group rather suggests that a new form of memory emerged during the sleep period, an effect known as "sleep-dependent memory evolution"[36] in adults.

## Relation between nap characteristics and newly evolved memory

The mid-latency negative memory effect of the nap group shares similarities with the N400 memory effect observed in the lexical-semantic learning studies in infants from 6 to 16 months of age. Based on evidence that the N400 memory effect in these previous studies depends on central-parietal sleep spindle activity[11,15,16] and, in 6- to 8-month-olds, also on the amount of stage 2 sleep in the post-encoding nap[15], in a next step we analysed the potentially relevant sleep characteristics (Supplementary Table 2) and their relationship to the N400-like memory effect after the nap.

In contrast to lexical-semantic learning at this age[15], the mid-latency N400-like memory effect of the nap group was not enhanced by more time spent in stage 2 sleep. On the contrary, greater amounts of stage 2 sleep tended to be associated with lower N400-like responses ($r = 0.253$, $P = 0.083$, 95% CI = [−0.034, 0.501]). This inverse association was even more pronounced for the duration of total NonREM sleep ($r = 0.383$, $P = 0.007$, 95% CI = [0.111, 0.601]). A closer look revealed that the correlation resulted from the retention of the old NAD phrases ($r = 0.401$, $P = 0.005$, 95% CI = [0.132, 0.615]), and was not found for the generalisation of the NADs to new verb stems ($r = 0.089$, $P = 0.547$, 95% CI = [−0.200, 0.364]).

Also distinct from lexical-semantic learning in the previous studies[11,15], the N400-like memory effect was not related to the amplitude of central-parietal sleep spindles ($r = 0.134$, $P = 0.362$, 95% CI = [−0.156, 0.403]). Instead, it was correlated with the amplitude of frontal sleep spindles ($r = 0.380$, $P = 0.008$, 95% CI = [0.108, 0.600]). However, the positive correlation with the negative effect indicates that higher spindle amplitudes were related to a weaker memory effect. This inverse relation between spindle activity and memory once more contrasts with previous findings on infant lexical-semantic learning, where the N400 memory effect was stronger with higher sleep spindle activity[11,15].

## Relation between frontal sleep spindles and specific memory

Detailed inspection of the inverse relation between sleep spindles and the mid-latency N400-like memory effect revealed that it was selectively driven by spindles occurring over the left frontal and mid-frontal regions. These were related to the memory effect for NAD phrases with old verb stems, but not to the generalisation effect for phrases with new verb stems (Supplementary Table 3). To further specify the relation between spindle activity and memory, we performed an exploratory analysis by splitting the nap group according to the infants' individual mean left- and mid-frontal spindle amplitudes. The resulting subgroups (hereafter termed high-spindle

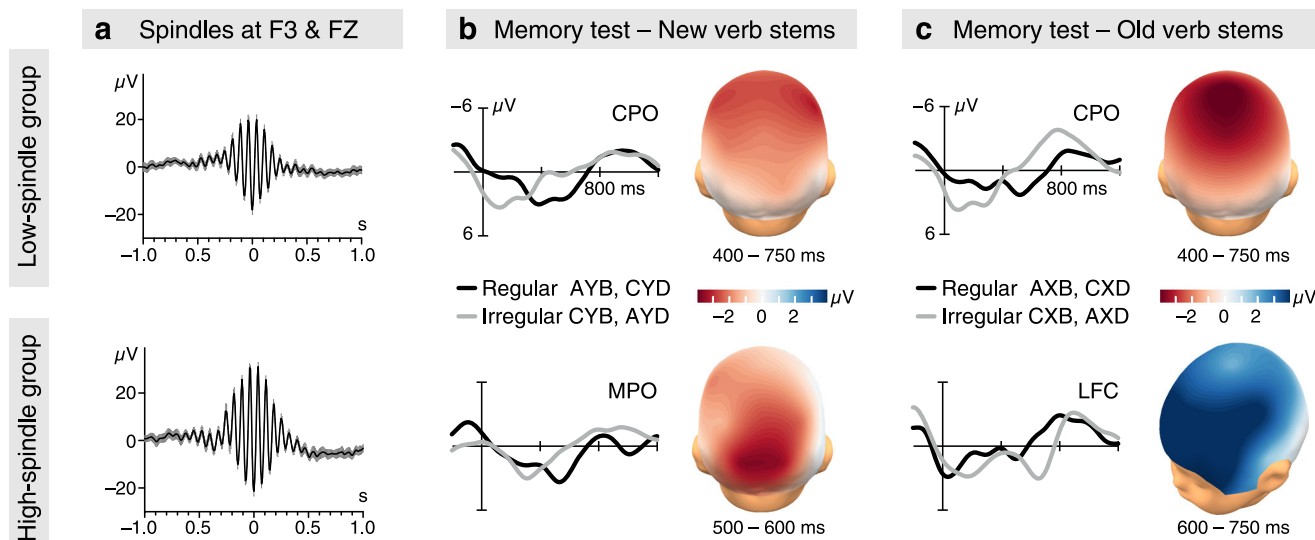

**Fig. 3 | Frontal sleep spindles and mid-latency memory effects in the spindle subgroups. a** Fast sleep spindles (mean ± SEM) during NonREM sleep of the post-encoding nap recorded over the left frontal (F3) and mid-frontal (FZ) brain regions in the two spindle subgroups, with mean peak-to-peak amplitudes of 35.83 μV (SD = 4.64) in the low-spindle subgroup (N = 24 infants) and 51.86 μV (SD = 6.86) in the high-spindle subgroup (N = 24 infants). For spindle number and spindle density, see Supplementary Table 4. **b** ERPs from suffixes in regular (black lines) and irregular (grey lines) NAD phrases with new verb stems and the N400-like memory

effect for new verb stems in the low- and high-spindle subgroups. CPO as in Fig. 2d. The mid-parietal-occipital (MPO) region included the PZ, O1, and O2 positions. Voltage maps as in 2b. **c** ERPs from suffixes in regular and irregular NAD phrases with old verb stems in the spindle subgroups, the N400-like memory effect for phrases with old verb stems in the low-spindle subgroup, and the left FCMR for phrases with old verb stems in the high spindle subgroup. CPO as in Fig. 2d. The left frontal-central (LFC) region included the F7, F3, FC3, and C3 positions. Voltage maps as in 2b. Source data are provided as a Source Data file.

subgroup and low-spindle subgroup) did not differ in age ($t_{46} = -0.490$, $P = 0.626$, 95% CI = [−16.269, 9.894], t-test for independent samples). Infants of the high-spindle subgroup had not only higher amplitudes (Fig. 3a), but also greater numbers and densities of spindles in the left frontal cortex (Supplementary Table 4). In the memory test phase, the subgroups expectedly differed in their mid-latency ERP responses of 400−600 ms (Regularity × Subgroup $F_{1,46} = 12.996$, $P = 0.001$, $\eta_p^2 = 0.220$; repeated measures ANOVA). However, the difference between subgroups was even more pronounced in the subsequent 600−750 ms time window of the ERPs (Regularity × Subgroup $F_{1,46} = 27.107$, $P = 0.000004$, $\eta_p^2 = 0.371$, Regularity × Region × Subgroup $F_{1,46} = 3.666$, $P = 0.033$, $\eta_p^2 = 0.074$, Old/New × Regularity × Subgroup $F_{1,46} = 6.582$, $P = 0.014$, $\eta_p^2 = 0.125$).

In the low-spindle subgroup, the central-to-occipital N400-like effect of the overall nap group was observed in the whole mid-latency time range of 400−750 ms ($t_{23} = -5.228$, $P = 0.00003$, $d = -1.067$, 95% CI = [−2.940, −1.274]). The effect was present for both old ($t_{23} = -4.220$, $P = 0.0003$, $d = -0.861$, 95% CI = [−3.469, −1.187]) and new ($t_{23} = -3.153$, $P = 0.004$, $d = -0.644$, 95% CI = [−3.125, −0.649]) verb stems (Fig. 3), with no significant difference between these separate effects ($t_{23} = -0.536$, $P = 0.597$, 95% CI = [−2.140, 1.259], t-test for dependent samples). This independence of the intervening verb stem indicates that the mid-latency ERP response of the low-spindle subgroup reflects a stage of generalised memory processing.

In contrast, the high-spindle subgroup differed in their responses to NAD phrases with old and new verb stems (Old/New × Regularity $F_{1,23} = 5.692$, $P = 0.026$, $\eta_p^2 = 0.198$, across the whole time window; Fig. 3). For new verb stems, a similar N400-like memory effect as in the low-spindle subgroup was detectable over the mid-parietal-occipital region (500−750 ms: $t_{23} = -2.506$, $P = 0.020$, $d = -0.511$, 95% CI = [−3.162, −0.302]), with a maximum within 500−600 ms (Fig. 3). However, for phrases with old verb stems, a pronounced polarity-inversed effect emerged within 600−750 ms, primarily over the left frontal ($t_{23} = 4.128$, $P = 0.0004$, $d = 0.843$, 95% CI = [2.066, 6.215]) and left central ($t_{23} = 4.107$, $P = 0.0004$, $d = 0.838$, 95% CI = [2.032, 6.157]) regions. To a somewhat lesser degree, the effect also involved mid-

frontal ($t_{23} = 3.225$, $P = 0.004$, $d = 0.658$, 95% CI = [1.313, 6.012]) and right frontal ($t_{23} = 3.077$, $P = 0.005$, $d = 0.628$, 95% CI = [1.108, 5.656]) regions. This special frontal-central memory response (FCMR) for old verb stems speaks for the existence of a memory that is specific to the individual old phrases and is not generalised to new phrases. The differential memory effects for phrases with old and new verb stems in the infants of the high-spindle subgroup further implies that these infants consolidated two types of memory during their nap, one for the individual verb phrases and one for the nonadjacent morphosyntactic regularities.

Confirming the analyses of the high- and low-spindle subgroups, in the overall nap group, the left FCMR for phrases with old verb stems was positively correlated with the amplitude of fast sleep spindles over the left frontal ($r = 0.563$, $P = 0.00003$, 95% CI = [0.332, 0.730]), mid-frontal ($r = 0.582$, $P = 0.00001$, 95% CI = [0.357, 0.743]), and left central ($r = 0.400$, $P = 0.005$, 95% CI = [0.131, 0.614]) regions (Fig. 4a). A similar relation was neither found for the corresponding left frontal-central ERP difference for new verb stems (Fig. 4b), nor for the N400-like generalisation effect to new verb stems ($r = 0.127$, $P = 0.389$, 95% CI = [−0.163, 0.397], for the mean of left frontal, mid frontal, and left central spindle amplitudes). The difference in the correlations of the left FCMR and the N400-like generalisation effect with mid-left frontal-central spindle activity (Meng, Rosenthal, & Rubin's[37,38] $z = 3.089$, $P = 0.002$, two-sided) moreover suggests that distinct mechanisms underlie the sleep-dependent consolidation of specific memory for individual speech phrases and the sleep-dependent evolvement of generalised memory for morphosyntactic regularities.

## Discussion
Memory for rule-based nonadjacent dependencies in speech is a core component of grammar learning. Here, we show that infants as young as six months are able to retain NADs in memory. Contrary to lexical-semantic learning at this age[15], post-encoding sleep did not prove necessary for infants' continued familiarity with the morphosyntactic regularities and their generalisation to novel verb stems. The different effect of sleep on memory for morphosyntactic dependencies in the

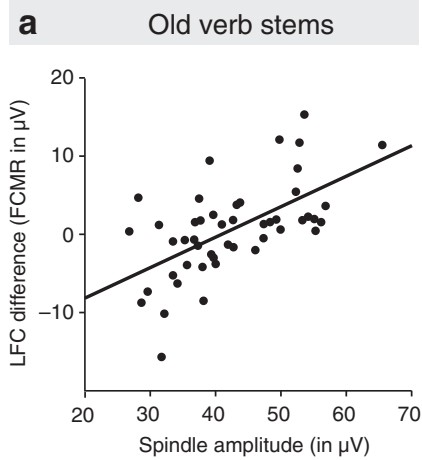
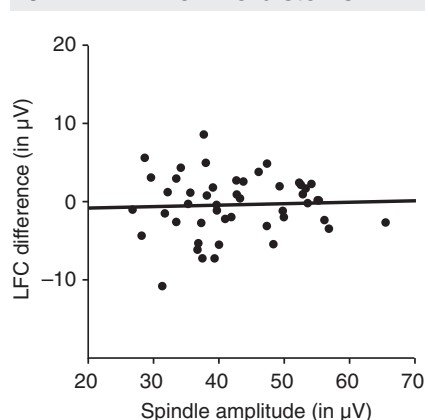
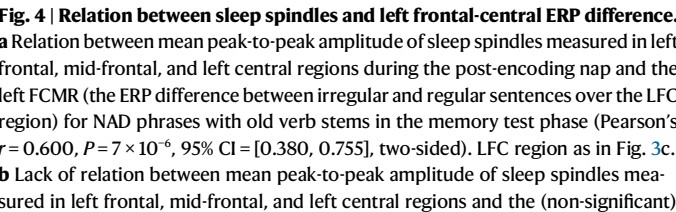

**Fig. 4 | Relation between sleep spindles and left frontal-central ERP difference.** **a** Relation between mean peak-to-peak amplitude of sleep spindles measured in left frontal, mid-frontal, and left central regions during the post-encoding nap and the left FCMR (the ERP difference between irregular and regular sentences over the LFC region) for NAD phrases with old verb stems in the memory test phase (Pearson's $r = 0.600$, $P = 7 \times 10^{-6}$, 95% CI = [0.380, 0.755], two-sided). LFC region as in Fig. 3c. **b** Lack of relation between mean peak-to-peak amplitude of sleep spindles measured in left frontal, mid-frontal, and left central regions and the (non-significant)

ERP difference between irregular and regular sentences in the LFC region for phrases with new verb stems (Pearson's $r = 0.045$, $P = 0.763$, 95% CI = [−0.243, 0.325], two-sided). The correlation coefficient for new verb stems significantly differed from that of old verb stems (Meng, Rosenthal, & Rubin's[37, 38] $z = −2.844$, $P = 0.004$, two-sided). For the corresponding relations between spindle RMS amplitude and left frontal-central ERP responses, see Supplementary Fig. 4. Source data are provided as a Source Data file.

infants in the present study and on memory for lexical-semantic dependencies in infants of the same age in a previous study[15] may indicate that the two types of memory do not share the same mechanisms of formation and consolidation. However, studies that directly compare the formation of syntactic and lexical-semantic memory are needed to further support the claim of a difference in these early memory mechanisms.

Despite the here observed general independence of sleep, a nap after the learning of morphosyntactic NADs altered a specific stage of subsequent NAD processing, which implies that sleep triggered further evolution of memory and changed the nature of its representations. Such a change in the nature of NAD memory, from representing surface statistics of sounds to that of higher-level morphological features, has been suggested to occur in children at an older age[39]. Here, the mid-latency memory effect of the wake group resembles not only the mid-latency familiarity effect of the encoding phase in the 6- to 8-month-olds of the present study, but also the ERP effect in younger infants, who heard the same sentences in a design with several alternating learning and test phases without a delay[7]. These similarities in the brain responses suggest that the memory effect of the wake group resulted from the encoding of surface statistics of sounds, such as the immediate formation of implicit associations of frequently occurring, temporally related auditory patterns.

The occurrence of the N400-like memory effect in the nap group points to more explicit, higher-level memory. Specifically, it suggests that the initially stored sound patterns were lexicalized during sleep. It might be the case that both the auxiliary/modal verb and the suffixes attained word status, and the order of these words was represented in memory. Alternatively, it could be speculated that the whole AXB and CXD verb phrases were stored as words, in which the middle part (X) was underspecified. In adults, violations of the expected order of such lexical or sub-lexical elements elicit an N400 response in the ERP, as known from studies on word fragment priming[40] and studies with varying levels of cloze probability[41,42]. Regardless of the type of lexicalization that might have occurred in the infants here, the presence of the N400-like effect only in the nap group strengthens the view that sleep promotes the formation of a more mature kind of memory that is usually first observed during later stages in development[15,39].

The N400-like memory effect of the infant nap group moreover resembles the N400 observed in adults, when they learn the same NADs of the present study in active designs with grammaticality judgement[27,43,44]. This learning process is affected by the activity of the prefrontal cortex (PFC), as can be inferred from the disappearance of the N400 in the test phase when PFC activity is down-regulated by cathodal transcranial direct current stimulation (tDCS) during the learning phase[44]. In particular, a positive generalisation effect emerged in adults after left PFC down-regulation, whereas under sham stimulation, the N400 was present. Notably, the adults' behavioural learning success was equal in these conditions, indicating that adults formed memory of the regularities both with and without PFC involvement, even though the learning mechanisms, and thus the type of memory, were altered by PFC activity. A similar distinction between memory in general and the specific type of representation might also account for the ERP pattern found in the infants here. The presence of two equivalent memory effects in the wake and nap groups, which, together with the mid-latency effect of the wake group strongly resemble the familiarity effects during encoding, evidences memory of NADs in both groups. The altered mid-latency memory effect in the nap group moreover suggests the evolution of a new type of memory that was neither established immediately during encoding nor formed in the infants of the wake group. The similarity of this infant N400-like response with the N400 adults displayed with functioning PFC may even indicate that, although largely immature, the infant PFC already contributes to memory processes in the first year of life. The emergence of a new type of memory in the infants of the nap group might therefore be an initial indication for an involvement of the maturing PFC in the sleep-dependent consolidation of infant memory.

Unlike previous studies on early NAD learning, in which either retention[25,26] or immediate generalisation[7] was tested, here we assessed both retention and generalisation within the same study. This design revealed that infants in the high-spindle subgroup had not only formed generalised memory of the nonadjacent regularities but also specific memory for the details of individual speech phrases. Like the specific memory of individual object-word pairs in 14- to 17-month-olds[22], the memory of individual speech phrases in the 6- to 8-month-olds here was related to spindle activity in the left frontal and adjacent brain regions. Also in both studies, the specific memories mainly

affected brain responses in the left frontal cortex, with only some delay in latency in the younger as compared to the older infants. Moreover, as with the old object-word pairs in the 14- to 17-month-olds, the 6- to 8-month-olds, who had consolidated memory for the individual speech phrases during their nap, preferentially accessed this specific memory when re-exposed to old phrases. These striking similarities in the memory responses in the two studies and their close relationship to the same left frontal sleep process argue for the existence of a shared mechanism of consolidating detailed memory. Whether this mechanism represents an early form of episodic binding or a kind of implicit associative or holistic memory formation remains to be clarified by future studies.

The present study has several limitations. One is that we could not predict the ERP components of interest a-priori, as there has been no ERP study investigating NAD memory in the first year of life. Our strategy to overcome this limitation was to use the ERP components indicating immediate memory formation in the encoding phase as a reference for the assessment of memory in the test phase. A further limitation is that at this stage we cannot rule out the possibility that individual traits or developmental differences between infants have contributed to the relation between left frontal sleep spindles and specific memory. In particular, we do not know whether the observed correlations reflect a direct involvement of sleep spindles in the consolidation process or whether infants with higher spindle activity in left frontal and adjacent cortex regions are generally more gifted in consolidating highly specific memories. This fundamental question requires further research and could be explored by within-subject studies that include a baseline assessment of spindle activity, as was done in a previous infant study that focused on the relationship between central-parietal spindles and lexical-semantic memory[16].

Despite these limitations, the present data suggest that a mechanism of consolidating specific memory of individual speech phrases is effectively established in infants at six months of age. This finding joins a large number of behavioural studies showing that early memory is indeed highly specific[45–47]. It further extends the existing knowledge in two ways. First, behavioural evidence of specific memory usually comes from the absence of flexible memory use, i.e., a lack of generalization. The present study, however, demonstrates detailed retention of individual speech patterns even despite the extraction of regularities and their generalization to novel verb stems. Second, in the behavioural studies, memory is commonly tested for only one or a very few different events. Here, infants were presented with 16 verb stems, occurring each four times in two different NAD phases in the encoding phase. Even though infants certainly did not retain all of these individual verb phrases, our data suggests that infants with high left frontal spindle amplitudes formed at least some memories with highly overlapping information – an ability, for which the hippocampus is thought to be responsible[48,49]. This finding challenges the notion that due to the immaturity of the hippocampus, infants in their first year of life are unable to form highly specific memories with overlapping information[50,51]. Given that functional activity of the immature hippocampus has recently been demonstrated for encoding regularities in the temporal occurrence of highly specific visual patterns in infants as young as 3 months of age[52], it is not entirely unlikely that the hippocampus was also involved in encoding and consolidating the highly specific, overlapping speech patterns in the 6- to 8-month-old infants of the present study.

Beyond the ability to process highly specific auditory information, language learning requires memory for word meanings and memory for how words are combined and modified according to the grammar of a language. The present study reports findings on infant memory for rule-based morphosyntactic dependencies. As early as 6 months, human infants are able to form memory for regularities in the co-occurrence of nonadjacent elements in sentences of an unknown natural language. We conclude that, in conjunction with the ability to transform generalised visual patterns into word meanings[15], the basic memory mechanisms of language-learning are set up at 6 months of age.

## Methods

### Participants

Data were obtained from 85 monolingual infants growing up in German-speaking families (41 female, mean age 7 months and 7 days, SD 24 days). An additional 40 infants (15 from the nap group, 25 from the wake group) were measured, but not included in the analyses because of: too few artefact-free trials or very noisy ERP responses due to fussiness or too much movement ($N = 23$), technical problems or loss of data ($N = 10$), break-off due to strong agitation or crying in one of the experimental sessions ($N = 3$), failure to fall asleep in the nap group ($N = 2$), or due to atypical sleep EEG ($N = 2$). Sample size was chosen based on previous studies[11,15,22], and thus no statistical method was used to predetermine sample size. All parents gave informed consent before participation. The study complied with all relevant ethical regulations and was approved by the ethics committee of the department of Psychology of the Humboldt University of Berlin.

All infants were born in the 36th to 43st week of pregnancy with a birth weight ranging from 2440 to 4900 g (mean: $3561 \pm 456$ g). They had no known hearing deficits and no major sleep problems. Prior to the experimental sessions, infants were assigned to either the wake group or the nap group. The infants were naturally blind to this experimental variation. However, the parents and the experimenter knew about the assignment to the groups, because the babies had to be scheduled according to their typical nap time. Infants of the nap group were scheduled at a time when they were expected to take a nap within 30 min after the encoding session. Infants of the wake group were scheduled at a time when they were expected not to take a nap within the next two to three hours. The nap group ($N = 48$, 25 female) and the wake group ($N = 37$, 16 female) neither differed in age, gestational age at birth, birth weight, or head circumference at birth (Supplementary Table 1) nor in Apgar scores at 10 min after birth (available in 76 of 85 infants, median: 10 in both groups, Mann-Whitney: $Z = -0.721$, $P = 0.471$).

### Stimuli

As stimulus material, we used sentences of the Italian miniature language, which had been created for studying NAD learning in 4-month-olds[7]. Sentences were composed of a noun phrase (*la sorella* or *il fratello*), followed by a verb phrase consisting of an auxiliary or a modal verb (*sta* or *può*, formalised as A or C), one of 32 verb stems (indicated as X or Y), and a suffix (*–ando* or *–are*, formalised as B or D) that depended on the (nonadjacent) auxiliary or modal verb (Fig. 1). We chose a set of sentences, in which a splicing procedure ensured that the prosodic patterns of the verb stem could not predict the suffix[27]. In order to control for acoustic-phonological differences, two versions of the language were created and the modal verb/auxiliary–suffix combinations were balanced between participants.

### Procedure

Infants participated in two experimental sessions, the encoding session and the memory test session, during which they passively listened to each 128 sentences of the miniature language. Both sessions were conducted on the same day, with a retention period of about 0.5 to 1.5 h between them. Sentences were presented with Presentation 17.2. (NeuroBehavioral Systems, Berkeley, USA). In order to minimise head movements and to increase compliance, infants looked at a silent baby movie while the sentences were presented via loudspeakers. Each session lasted for approximately 10 min.

In the encoding session, each infant was familiarised with one version of the language. Infants heard half of the sentences of their language twice. To avoid any interference with the learning of adjacent

dependencies, each of the 16 verb stems was presented in both NAD phrases, and thus, were paired with both suffixes (e.g., *cant–ando*, *cant–are*). An individual phrase with a specific verb stem (e.g., *sta cant–ando*) occurred four times.

In the retention period, one group of infants napped and another group stayed awake. After the encoding session, infants of the nap group were prepared for polysomnographic recordings and were laid down in a baby crib or pram, or were held by their parent until they fell asleep. On average, infants napped for 39.6 min (SD 17.8 min). Compared to the retention time of the nap group ($73.2 \pm 21.9$ min), we shortened the retention time of the wake group ($37.8 \pm 8.8$ min) in order to ensure that infants stay alert during the memory test. As a result, the mean wake retention time in the nap group ($33.6 \pm 12.5$ min) did not significantly differ from the mean retention time in the wake group ($t_{83} = 1.731$, $P = 0.087$, 95% CI = [−8.99, 0.62], *t*-test for independent samples). Also, the time of the day at which the encoding session was applied ($11:29 \pm 1:45$ h for the end of the encoding session) did not differ between groups ($t_{83} = −0.184$, $P = 0.854$, 95% CI = [−0:50, 0:41], wake group: $11:28 \pm 1:39$ h, nap group: $11:32 \pm 1.55$ h).

In the memory test session after the retention period, infants were exposed to sentences of both versions. In particular, they heard the 64 regular sentences of their language version with the familiarised NADs of the encoding session and 64 irregular sentences of the other version, which violated the familiarised NADs. Half of the sentences of each version contained an old verb stem that had already been presented during encoding (similar as in Gomez and colleagues[25,26]) and the other half contained a new verb stem that was not previously presented (as in Friederici and colleagues[7,27,43,44]).

## ERP data acquisition and analyses

During the encoding and memory test sessions, the EEG was recorded with a stationary system (XREFA with QXREFA 82 software, Twente Medical Systems International, Oldenzaal, The Netherlands) at 21 electrode sites and digitised on-line at a rate of 500 Hz. Offline, the EEG was re-referenced to the average of left and right mastoids. EEG preprocessing was done with EEP 3.2.1. (MPI for Human Cognitive and Brain Sciences, Leipzig, Germany). A zero-phase digital band-pass filter ranging from 0.5 to 20 Hz (−3 dB cut-off frequencies at 0.61 and 19.89 Hz) was applied. As in previous studies[16,22], the strong DC-suppression of this filter (−90 dB) enabled the calculation of ERPs without baseline correction. ERPs were averaged time-locked to the onset of the suffix. Trials exceeding a standard deviation of 90 µV within a sliding window of 500 ms at any electrode site were rejected. A minimum of 10 artefact-free trials per condition was defined as required for the inclusion of an individual in further analyses. For the encoding session, the mean number of trials was 41 (SD = 10) for the initial presentation of the 64 sentences in the first half, and 35 (SD = 12) for the repetition of the sentences in the second half. In the memory test session, on average 22 (of 32) trials (SD = 4) per condition contributed to an individual participant's ERP. Trial numbers did not differ between conditions ($F_{3,252} = 0.873$, $P = 0.452$; repeated measure ANOVA). Also, the mean number of trials did not differ between groups, neither for the encoding session ($t_{83} = 0.515$, $P = 0.608$, *t*-test for independent samples) nor for the memory test session ($t_{83} = 1.125$, $P = 0.264$).

For the statistical analyses we used SPSS 22 and SPSS 28 (IBM, Armonk, USA). In order to increase statistical power, regions of interest (ROIs) were defined by averaging the ERP responses from each two lateral recording sites: F7 and F3 formed the left frontal region (LF), F8 and F4 the right frontal region (RF), C3 and FC3 the left central region (LC), C4 and FC4 the right central region (RC), P3 and CP5 the left parietal region (LP), and P4 and CP6 the right parietal region (RP). FZ was taken as the mid-frontal region (MF), CZ as the mid-central region (MC), and PZ, O1, and O2 were included in the mid-parietal-occipital region (MPO). After visual inspection of the ERPs, we analysed three time windows relative to suffix onset: early (100–300 ms), middle

(400–600 ms), and late (encoding: 700–850 ms, memory test: 750–900 ms). For the explorative analyses of the spindle subgroups, we analysed an additional time window with an intermediate latency (600–750 ms). For better illustration of the effects, a low-pass filter of 7 Hz was applied to the ERPs shown in Figs. 2, 3 and in the Supplementary Figs. 1–3.

In order to assess the immediate formation of memories during familiarisation, we performed within-subjects ANOVAs on the ERP data of the encoding session with three factors: Familiarity (first half vs. second half), Laterality (left, mid, and right), and Region (frontal, central, and parietal-occipital). To evaluate retention, generalisation, and their dependency on sleep, we conducted ANOVAs on the data of the memory test phase with the within-subject factors Regularity (regular structure vs. irregular structure), Old/New (old verb stems vs. new verb stems), Laterality, and Region as well as the between-subject factor Group (wake vs. nap). Normality was assessed by Q-Q-diagrams. In order to compensate for any deviations from sphericity, in all ANOVAS, Greenhouse–Geisser corrections were applied. To capture possible differences between male and female infants, all ANOVAs were repeated with an additional between-subject factor sex (female/male). These analyses yielded the same results and revealed no influence of sex on memory. For post-hoc tests, illustrations, and correlations, averaged ERP differences were calculated for the frontal, central, and parietal-occipital regions. ERP responses of the frontal region included the LF, MF, and RF ROIs and were calculated by averaging the responses at the electrode positions F7, F3, FZ, F4, and F8. Those of the central region included the LC, MC, and RC ROIs and were calculated as the average of FC3, C3, CZ, FC4, and C4. The parietal-occipital region was composed of the LP, RP, and MPO ROIs by averaging P3, CP5, P4, CP6, PZ, O1, and O2. The central-to-occipital (CPO) region included the central and mid-parietal-occipital regions and was calculated as the mean of the electrode positions FC3, FC4, C3, CZ, C4, P3, PZ, P4, CP5, CP6, O1, and O2. Prior to running a *t*-test, normality was tested using Kolmogorov-Smirnov. In one case where normality was violated, the result of the *t*-test was confirmed by an additional non-parametric test (Wilcoxon). For multiple testing at two regions, a conservatively adjusted significance level of 0.025 was chosen to correct for multiple comparisons. When testing three regions, we used a significance level of 0.017, for the parallel testing of nine ROIs, 0.006 was chosen. When testing differences within groups, Cohen's d was calculated as the quotient of the mean and the standard deviation of the differences. When applying *t*-tests for independent samples, Cohen's d was calculated as the quotient of the mean group difference and the pooled standard deviation. For *t*-tests, the 95% confidence interval (95% CI) refers to the mean difference tested. All statistical tests were two-sided.

## Sleep recordings and sleep spindle analyses

Sleep was recorded using a portable amplifier (SOMNOscreen EEG 10–20, DOMINO Steuerung 2.5., Somnomedics, Randersacker, Germany). EEG recordings were obtained with electrodes attached at F3, FZ, F4, C3, C4, P3, PZ, and P4 as well as left and right mastoids, referenced to CZ filtered between 0.03 and 35 Hz, and sampled at 256 Hz. The electrooculogram and the electromyogram were recorded bipolarly from electrodes close to the eyes and under the chin, respectively. Offline, EEG signals were re-referenced to the average potential at the left and right mastoid electrodes. Sleep recordings were visually scored according to standard criteria[53]. For each nap, total sleep time and the time spent in the different sleep stages (1, 2, slow wave sleep, and REM sleep) were determined. Sleep scoring was performed using REMBRANDT 9 Diagnostic software (Natus Medical, Pleasanton, USA).

The algorithm for the detection of discrete sleep spindles was the same as in our previous infant studies[11,15,16,22]. It was written with SPIKE2 9.12 (Cambridge Electronic Design, Milton, Cambridge, United Kingdom). First, for each infant and each channel, the individual peak

frequency of fast sleep spindles ($13.65 \pm 0.45$ Hz, across all infants and channels) was identified in the EEG power spectra of the low-pass filtered (32 Hz) and down-sampled (128 Hz) EEG of all artefact free NonREM epochs. The EEG signal was then band-pass filtered with a width of 3 Hz centred on the detected individual peak frequency. A root mean-square (RMS) representation of the filtered signal was calculated and smoothed using a sliding window of 0.2 s with a step size of one sample. Time frames were considered as spindle intervals if the RMS signal exceeded a threshold of 1.5 standard deviations of the filtered signal for 0.5–5 s. Two succeeding spindles were counted as one spindle when the interval between the end of the first spindle and the beginning of the second spindle was shorter than 0.5 s and the resulting (merged) spindle was not longer than 5 s. The mean peak-to-peak amplitude of fast spindles was calculated in each infant for each channel by averaging the peak-to-peak amplitude of all detected individual spindles. Spindle amplitudes were averaged across each of the three individual electrode positions to obtain the values for frontal (F3, FZ, F4), central (C3, CZ, C4), and parietal (P3, PZ, and P4) regions.

### Reporting summary

Further information on research design is available in the Nature Portfolio Reporting Summary linked to this article.

## Data availability

The data that support the findings of this study have been deposited in the OSF repository and are available at https://osf.io/q2vpg/. Source data are provided with this paper.

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

## Acknowledgements
We thank all families who participated in this study. Special thanks to Francesca Citron for sharing the experimental sentences, to Franziska Illner for specifying the experimental set-up, to Benjamin Schlütter for providing the presentation programs, to Kerstin Strelow-Morgenstern for scoring sleep stages, and to Jördis Haselow and Christina Rügen for recording the infant EEG data. The study was supported by grants from the Deutsche Forschungsgemeinschaft to M.F. (DFG 222228420 and 409092104) and a grant from the European Research Council to J.B. (ERC AdG 883098 SleepBalance).

## Author contributions
Conceptualization, M.F.; Analysis, M.F. and M.M.; Writing – Original Draft, M.F.; Writing – Review & Editing, M.M., J.B. and A.D.F.; Resources, M.F. and A.D.F.; Funding Acquisition, M.F.

## Funding

## Competing interests
The authors declare no competing interests.
