## [Peer Review File · Nature Communications]

Memory for nonadjacent dependencies in the first year of life and its relation to sleepReviewers' comments:

Reviewer #1 (Remarks to the Author):

Existing behavioural evidence suggests that sleep modulates memory for non-adjacent dependencies in 15 month olds. However, this is yet to be examined in younger infants. Thus, the present study utilises event-related potentials (ERPs) to assess the learning and memory of NADs in 6- to 8-month-olds, as well as recording naps with polysomnography during the time that elapsed between learning and test, to examine the relationship between sleep and morphosyntactic memory (drawing comparisons with previous findings from lexical-semantic memory). Infants were exposed to NADs in a familiarisation phase, and then either napped or stayed awake, before having their ERPs recorded. At the ERP test phase, infants heard regular sentences with the familiarised NADs (A–B and C–D) as well as irregular sentences that violated the familiarised structure (A–D and C–B). Half of the sentences contained an old intermediate verb stem of the encoding phase (X) to test for retention, the other half a new verb stem (Y) to test for generalisation. The results are clear in suggesting that retention and generalisation of NADs occurs over wake and a daytime nap, but that some elements of learning (specifically, memory for individual speech strings) are sleep-dependent. The findings that arise from the exploratory analyses that compare low and high frontal spindle groups are particularly neat, and clearly demonstrate a differentiation of sleep effects for individual speech string memory, versus memory for NADs. This is a novel and clean design with a large sample size for a study of this type, that well addresses the research questions posed. There is clear evidence of theoretical innovation and a novel contribution to the literature. My comments mainly focus on improving the clarity of the set up of the questions and the conveying of the key findings.

You refer to “sensitivity” to NADs as different to “memory” for NADs. Can you clarify how you distinguish these, both conceptually/neurocognitively and in their measurement? There’s a general lack of clarity with a few different terms. This issue first crops up in the abstract e.g., “Results also challenge the notion that immature brain structures prevent the formation of precise memories in the first year of life.” What’s a “precise” memory? And, “napping modified the nature of the memory” - can you replace “nature” with a more specific term?

Summary of the key results on page 86. I found mapping the key effects on to the design outlined in the previous paragraph to be a little difficult, you can certainly make life easier for the reader at this point. For example, you begin by stating that generalisation didn’t depend on sleep without first clarifying whether familiarity of the NADs depended on sleep. You then refer to the relationship between spindle activity and memory for “individual speech phrases”, but it’s not clear what kind of memory you’re referring to here. Furthermore, you mention in the abstract that “napping, however, modified the nature of memory” which I was particularly keen to read more about in this paragraph, but it didn’t appear. Finally, at this point, it’s not clear whether the comparisons with lexical-semantic memory being underpinned by different mechanisms to NAD learning come purely from comparisons with previous findings, or whether the present data directly speak to that claim. (On that point, there’s a danger with drawing comparisons with previous datasets that differ methodologically given that one variable that we know is a strong predictor of whether expected sleep effects in language learning emerge is encoding performance / task difficulty, thus seeing a sleep effect in one experiment but not in another doesn’t necessarily point to “neural mechanisms that are distinct from those observed for semantic memory formation”). Generally then, clarifying which contrasts test which aspects of memory and which aspects are sleep-dependent and which aspects are not, needs tightening up by this point.

I found the blanket subheading “Retention and generalisation of NADs does not depend on sleep” to be a little misleading given there was an effect of wake/nap for the mid-latency ERP. Also, whilst there’s some hint of what the early and late effects might reflect in the Results section (on page 6), there’s very little hint of what the mid-latency ERP might reflect at this point (this doesn’t come until later when you liken it to the N400 ERP), and thus referring to it as a marker of “reorganisation” seems like a big leap. Generally, my unease with these subheadings reflects that the ERPs of focus have not yet been set up in an apriori way, and that the Results section (particularly for the retention

aspects) feels like it's structured around the findings rather than around clear hypotheses that map on to each of the ERP segments.

One key limitation of the design is that the total retention period was significantly longer for the nap group than the wake group (despite the time awake in both groups being equated). This should be flagged, and where relevant, it might be helpful to address this limitation by examining whether the mean effect over regions demonstrating the mid-latency negativity was associated to the time between familiarisation and test.

Reviewer #2 (Remarks to the Author):

This manuscript reports the results of an experiment (N = 85) on post-encoding sleep-effects on 6- to 8-month-old infants' memory in the context of language learning. In contrast to earlier findings on the role of sleep for lexical-semantic learning, there was no effect of sleep on retention (and generalization) of nonadjacent dependencies (NADs) – both infants who napped and who stayed awake during the delay evidenced retention. Analyses of the mid-latency ERP segment revealed a similarity of processing of the NADs during encoding and test in infants in the wake group. Infants in the nap group showed a different mid-latency memory effect at test. There was a negative correlation between length of stage 2 sleep and NREM sleep in total and the mid-latency response. Frontal sleep spindle amplitude was negatively related to the strength of the memory effect (again, different from prior findings on lexical-semantic language learning). Follow-up analyses using a median split (to distinguish between infants with low vs. high spindle amplitudes) revealed that brain responses in the low spindle group did not differ between sentences with old vs. new word stems whereas responses in the high spindle group did.

These are intriguing results from a well-designed experiment. The methodological approach is thorough. Attrition rates seem comparable to similar studies. I wondered about one aspect of the procedure. Understandably, the authors provided infants with a distraction during the listening procedures at encoding and test that is, infants watched a baby movie. Given that the authors address the issue of declarative/declarative-like vs. nondeclarative memory in their manuscript I wondered how much they think that infants' attention went into the listening task. Assessing "conscious" recall/processing is a vexed issue in nonverbal populations. Nevertheless, perhaps the authors could speculate as to how much of the processing responses at test was conscious/declarative, or how this could be assessed in the future.

The analytic approach appears systematic and appropriate. The findings extend previous work by the authors and others on the role of sleep for memory retention and "evolution" (Stickgold & Walker, 2013) that is, qualitative changes in representations. They make a valuable contribution to the emerging field of experimental sleep research with infant samples. The findings also speak to questions around memory and brain development more generally by suggesting that infants can form precise memories at an age when the hippocampus is still immature.

The authors argue that this study shows that different aspects of language processing (lexical-semantic vs. memory for morphosyntactic dependencies) might be influenced by sleep in different ways. While this is an interesting possibility, it should be noted that the results regarding sleep characteristics and ERP indices are correlational, leaving open questions with regard to cause and effect (e.g., How can it be ruled out that mnemonically advanced infants simply also show specific sleep features?).

From my understanding, this study was not preregistered. Hence, it would be great if the authors could state in more places which of their analyses were confirmatory and which were exploratory in nature.

In the discussion (and perhaps in the introduction), the authors might consider contextualizing their

study a bit more. There is a rich developmental literature on early memory that has revealed the high specificity of infant memory (e.g., [https://doi.org/10.1016/0163-6383\(86\)90018-4](https://doi.org/10.1016/0163-6383(86)90018-4) ; <https://doi.org/10.1037/0735-7044.114.1.77> ; [https://doi.org/10.1002/\(SICI\)1098-2302\(199807\)33:1<61::AID-DEV6>3.0.CO;2-Q](https://doi.org/10.1002/(SICI)1098-2302(199807)33:1<61::AID-DEV6>3.0.CO;2-Q). Gradual reduction of specificity of memory retrieval is characteristic of early memory development. While this feature of early memory could be regarded as a deficit as it precludes effective generalization, it also shows that infants do indeed have highly specific memories. This way, this (slightly older) literature would support the authors' notion that formation of specific (or "precise") memories is a feature of very early cognition.

Reviewer #3 (Remarks to the Author):

The current manuscript reports the result of one ERP study examine whether infants ages 6- to 8-months retain memory for grammar, and whether post-encoding sleep affects the nature of memory representations. There is a lot to like about the manuscript including the questions beings addressed, the experimental design which is elegant and well controlled, and the focus on neural mechanisms.

The scope of this research, however, is limited by the lack of a behavioral measure of memory retention and generalization (e.g., head-turn paradigm, novelty preference paradigm). The authors correctly state that little is known about whether infants can retain information about grammatical structure and whether sleep modifies the nature of memory representations. Although the reported ERP findings suggest that memory is indeed retained and some generalization occurs, the conclusions are entirely based on the correspondence between the current findings and patterns of event-related potentials gleaned primarily from older groups. Without direct behavioral knowledge, it is entirely possible that these ERP findings do not quite correspond to the same phenomenon in young infants as they do in older infants and adults or do not manifest in overt behaviors that reflect retention. If the latter is the case, then we would have to ask what these pattern of neural activation correspond to and what they afford in terms of memory skill.

I am aware that equating ERP evidence to a cognitive skill, including memory, is not uncommon in the field of infancy. This limitation is typically addressed by exposing infants to experimental conditions that --it is argued--are constrained enough to reveal the predicted skill. However, this approach is not ideal when the fundamental behavioral outcome has not been established in the field (retention of grammar after a delay) and when there is ongoing debate on how to distinguish relevant memory constructs and pinpoint their emergence in early development (i.e., retention of specific details versus retention of general memories; e.g., Kerezstes et al., 2018; Ramsaran et al., 2019). With behavioral data demonstrating a pattern of results consistent with the ERP findings, this manuscript would be more likely to meet the standards of such a high profile journal.

Reviewer #1 (Remarks to the Author):

Existing behavioural evidence suggests that sleep modules memory for non-adjacent dependencies in 15 month olds. However, this is yet to be examined in younger infants. Thus, the present study utilises event-related potentials (ERPs) to assess the learning and memory of NADs in 6- to 8-month-olds, as well as recording naps with polysomnography during the time that elapsed between learning and test, to examine the relationship between sleep and morphosyntactic memory (drawing comparisons with previous findings from lexical-semantic memory). Infants were exposed to NADs in a familiarisation phase, and then either napped or stayed awake, before having their ERPs recorded. At the ERP test phase, infants heard regular sentences with the familiarised NADs (A–B and C–D) as well as irregular sentences that violated the familiarised structure (A–D and C–B). Half of the sentences contained an old intermediate verb stem of the encoding phase (X) to test for retention, the other half a new verb stem (Y) to test for generalisation. The results are clear in suggesting that retention and generalisation of NADs occurs over wake and a daytime nap, but that some elements of learning (specifically, memory for individual speech strings) are sleep-dependent. The findings that arise from the exploratory analyses that compare low and high frontal spindle groups are particularly neat, and clearly demonstrate a differentiation of sleep effects for individual speech string memory, versus memory for NADs. This is a novel and clean design with a large sample size for a study of this type, that well addresses the research questions posed. There is clear evidence of theoretical innovation and a novel contribution to the literature. My comments mainly focus on improving the clarity of the set up of the questions and the conveying of the key findings.

Authors' response: We thank the reviewer very much for her/his appreciation.

(1) You refer to “sensitivity” to NADs as different to “memory” for NADs. Can you clarify how you distinguish these, both conceptually/neurocognitively and in their measurement?

Authors' response: By “sensitivity”, we mean the unconscious immediate detection of regularities during encoding. To be more precise, we have replaced “sensitivity to NADs” by “encoding of NADs”.

(2) There’s a general lack of clarity with a few different terms. This issue first crops up in the abstract e.g., “Results also challenge the notion that immature brain structures prevent the formation of precise memories in the first year of life.” What’s a “precise” memory?

Authors' response: By "precise memory", we mean highly specific memory. We have replaced this phrase throughout the text.

(3) And, “napping modified the nature of the memory” - can you replace “nature” with a more specific term?

Authors' response: In response to the reviewers' comment, we have specified “napping modified the nature of memory” in the abstract by “During sleep, a new form of generalised memory evolved from initially represented dependencies between speech sounds” and further discussed the type of memory evolution at page 13.

(4) Summary of the key results on page 86. I found mapping the key effects on to the design outlined in the previous paragraph to be a little difficult, you can certainly make life easier for the reader at

this point. For example, you begin by stating that generalisation didn't depend on sleep without first clarifying whether familiarity of the NADs depended on sleep. You then refer to the relationship between spindle activity and memory for "individual speech phrases", but it's not clear what kind of memory you're referring to here.

Authors' response: We have rephrased the whole summary (p4-5):

"We provide clear evidence that 6- to 8-month-old infants retain their familiarity with NADs in memory and generalise the nonadjacent regularities to novel verb stems. In contrast to lexical-semantic memory examined in previous infant studies^{11,15}, sleep was not crucial for the memory of NADs. However, sleep after encoding modified subsequent NAD processing, suggesting that memory of the regularities further evolved during napping. Also different from lexical-semantic memory in the previous infant studies^{11,15,16}, the generalised memory for the morphosyntactic dependencies was not affected by sleep spindle activity during the nap. Our results imply that from the earliest stages of development, the extraction and storage of syntactic regularities relies on neural mechanisms that are distinct from those of semantic memory. However, similar to the detailed memory of individual object-word pairs in a recent study with older infants²², high spindle activity in the left frontal cortex was related to detailed memory of the old speech phrases presented in the encoding phase. From this, we conclude that a sleep-dependent mechanism of consolidating highly specific memory is effectively established within the first half year of life."

(5) Furthermore, you mention in the abstract that "napping, however, modified the nature of memory" which I was particularly keen to read more about in this paragraph, but it didn't appear. Finally, at this point, it's not clear whether the comparisons with lexical-semantic memory being underpinned by different mechanisms to NAD learning come purely from comparisons with previous findings, or whether the present data directly speak to that claim. (On that point, there's a danger with drawing comparisons with previous datasets that differ methodologically given that one variable that we know is a strong predictor of whether expected sleep effects in language learning emerge is encoding performance / task difficulty, thus seeing a sleep effect in one experiment but not in another doesn't necessarily point to "neural mechanisms that are distinct from those observed for semantic memory formation").

Authors' response: We have now already provided some more specific information about the change in the type of memory in the abstract (see our response to comment #3) and have addressed this issue in detail in the results and discussion sections.

We agree with the reviewer that seeing a sleep effect in one experiment but not in another does not necessarily point to distinct neural mechanisms. Here however, we found a very similar effect of sleep in both experiments, namely the emergence of an N400-like ERP difference in the memory test in the nap groups of both the present syntactic and the previous semantic study (while in both studies, a comparable effect was neither present during encoding nor in the memory test of the wake groups). At first glance, this suggests the presence of similar consolidation mechanisms in the two studies. However, we found that the relationship between these apparently same memory effects and the characteristics of the nap was quite different in the studies, indicating that although sleep appears to have a similar effect in both studies, the sleep mechanisms involved were not the same. In response to the reviewer's note, we have removed the comparison with lexical-semantic memory from the summary and focused more on the other two main findings (i.e. the retention and generalisation of NADs and the presence of highly specific speech memory), but continue to include

this comparison in the summary of main findings. In addition, we have carefully revised the paper and, where missing, have explicitly stated that the implication of different mechanisms underlying syntactic and lexical-semantic memory is based on comparisons with previous results.

“In contrast to lexical-semantic memory examined in previous infant studies^{11,15}, sleep was not crucial for the memory of NADs. ... Also different from lexical-semantic memory in the previous infant studies^{11,15,16}, the generalised memory for the morphosyntactic dependencies was not affected by sleep spindle activity during the nap. ... However, similar to the detailed memory of individual object-word pairs in a recent study with older infants²², high spindle activity in the left frontal cortex was related to detailed memory of the old speech phrases presented in the encoding phase.” (p4-5)

“This stands in contrast to the memory for object-word relations previously observed at this age¹⁵ and suggests a difference in the memory mechanisms underlying lexical-semantic and syntactic dependency learning.” (p7)

“The differential effects of sleep on the generalisation of morphosyntactic dependencies to novel verb stems in the infants in the present study and on the generalisation of lexical-semantic dependencies to novel objects in infants of the same age in a previous study¹⁵ suggests that the two types of memory do not share the same mechanisms of formation and consolidation.” (p12)

(6) Generally then, clarifying which contrasts test which aspects of memory and which aspects are sleep-dependent and which aspects are not, needs tightening up by this point.

Authors' response: We have clarified this in the summary. Please see the new summary in our response to comment #4.

(7) I found the blanket subheading “Retention and generalisation of NADs does not depend on sleep” to be a little misleading given there was an effect of wake/nap for the mid-latency ERP. Also, whilst there’s some hint of what the early and late effects might reflect in the Results section (on page 6), there’s very little hint of what the mid-latency ERP might reflect at this point (this doesn’t come until later when you liken it to the N400 ERP), and thus referring to it as a marker of “reorganisation” seems like a big leap.

Authors' response: We have changed the subheading into “Same memory effects indicate retention and generalization in the wake and nap groups”. In this paragraph, we did not refer to the mid-latency effect, because there was no main effect in the overall group.

We have also specified the next paragraph, and (in response to Reviewer#2’s comment #2) now use the terms “evolvment” or “evolution” instead of “reorganization” when describing the emergence of the qualitatively new memory effect in the nap group:

“Compared to the processing of NADs during encoding, this mid-latency memory effect of the nap group represents a qualitative shift in a particular processing stage, implying a modification of the underlying memory representations during the retention period. In the wake group, individual retention time was not correlated with the respective ERP difference ($r = .103$, $P = .542$), ruling out the possibility that the longer mean retention time in the nap group was responsible for the modification of memory. This suggests that the memory qualitatively changed in the sleep period,

and thus, some “memory evolution”³⁶ occurred during the nap.”(p9)

(8) Generally, my unease with these subheadings reflects that the ERPs of focus have not yet been set up in an apriori way, and that the Results section (particularly for the retention aspects) feels like it's structured around the findings rather than around clear hypotheses that map on to each of the ERP segments.

Authors' response: Unfortunately, to date there exists only one ERP study on NAD processing in the first year of life (Friederici et al., 2011, in 4-month-olds) that also did not involve memory. Although we hypothesized that infants are capable of forming NAD memory, and we expected that sleep would affect this memory, which should be reflected in ERP differences between wake and nap groups, we could not predict the corresponding ERP components. Our strategy to overcome this limitation was to use the familiarity responses from the encoding phase as a reference for memory assessment.

In response to the reviewers' comment, we have now changed the subheadings to:

“Same familiarity effects indicate equivalent encoding in the wake and nap groups” (p6),

“Same memory effects indicate retention and generalization in the wake and nap groups” (p7),

“New memory effect in the nap group indicates evolvment of memory during sleep” (p8),

“Sleep mechanisms of consolidation differ between syntactic and semantic memory” (p9),

“Frontal sleep spindles are related to highly specific memory” (p10).

(9) One key limitation of the design is that the total retention period was significantly longer for the nap group than the wake group (despite the time awake in both groups being equated). This should be flagged, and where relevant, it might be helpful to address this limitation by examining whether the mean effect over regions demonstrating the mid-latency negativity was associated to the time between familiarisation and test.

Authors' response: Unfortunately, the time between familiarisation and test strongly depends on the duration of the nap, for which we reported a correlation with the mid-latency negativity in the nap group. In order to flag and clarify this issue, we have now analyzed the relation between retention time and the mid-latency effect (of the nap group) in the wake group and included the following sentence:

“In the wake group, individual retention time was not correlated with the respective ERP difference ($r = .103$, $P = .542$), ruling out the possibility that the longer mean retention time in the nap group was responsible for the modification of memory.” (p9)

Reviewer #2 (Remarks to the Author):

This manuscript reports the results of an experiment (N = 85) on post-encoding sleep-effects on 6- to 8-month-old infants' memory in the context of language learning. In contrast to earlier findings on the role of sleep for lexical-semantic learning, there was no effect of sleep on retention (and generalization) of nonadjacent dependencies (NADs) – both infants who napped and who stayed awake during the delay evidenced retention. Analyses of the mid-latency ERP segment revealed a similarity of processing of the NADs during encoding and test in infants in the wake group. Infants in the nap group showed a different mid-latency memory effect at test. There was a negative correlation between length of stage 2 sleep and NREM sleep in total and the mid-latency response. Frontal sleep spindle amplitude was negatively related to the strength of the memory effect (again, different from prior findings on lexical-semantic language learning). Follow-up analyses using a median split (to distinguish between infants with low vs. high spindle amplitudes) revealed that brain responses in the low spindle group did not differ between sentences with old vs. new word stems whereas responses in the high spindle group did.

These are intriguing results from a well-designed experiment. The methodological approach is thorough. Attrition rates seem comparable to similar studies.

Authors' response: We thank the reviewer for her/his appreciation.

(1) I wondered about one aspect of the procedure. Understandably, the authors provided infants with a distraction during the listening procedures at encoding and test that is, infants watched a baby movie. Given that the authors address the issue of declarative/declarative-like vs. nondeclarative memory in their manuscript I wondered how much they think that infants' attention went into the listening task. Assessing "conscious" recall/processing is a vexed issue in nonverbal populations. Nevertheless, perhaps the authors could speculate as to how much of the processing responses at test was conscious/declarative, or how this could be assessed in the future.

Authors' response: Our study was intended to capture grammar memory, which is primarily non-declarative (but may become conscious to some extent). In addition, we aimed to make the design comparable to that of Friederici et al. (2011), who found that, while watching a baby movie, 4-month-old infants are able to encode the same NADs as we used in our study. We agree with the reviewer that infants might have directed their attention toward listening, but we cannot quantify how much attention they paid to the sentences. The comparison of the mid-latency brain response of our infant nap group with that of adults in an attention-demanding task with the same stimuli could possibly indicate that sleep has transferred non-declarative to declarative memory or has triggered some awareness of the grammatical errors in the memory test in these infants (although we would hesitate to use the term "declarative" here). This would fit with findings in adults that sleep provides insight into hidden rules and makes non-declarative memory consciously accessible. However, because we cannot measure an infant's awareness, we will not speculate on this (certainly exciting) question in the present manuscript.

Nevertheless, we discussed the sleep-dependent evolution of a more mature form of memory for the morphosyntactic regularities (pages 12-14) and a possible involvement of the hippocampus in consolidating highly specific memory in the infants of the high-spindle subgroup (see our response to comment #5).

(2) The analytic approach appears systematic and appropriate. The findings extend previous work by the authors and others on the role of sleep for memory retention and “evolution” that is, qualitative changes in representations. They make a valuable contribution to the emerging field of experimental sleep research with infant samples. The findings also speak to questions around memory and brain development more generally by suggesting that infants can form precise memories at an age when the hippocampus is still immature.

Authors’ response: Thanks again for this appreciation and for suggesting the term “evolution”, which we have now used (along with the somewhat more cautious term "evolvement") to describe the sleep-dependent change in memory, as our previous term "reorganisation" rather suggests that the originally formed memory is lost.

(3) The authors argue that this study shows that different aspects of language processing (lexical-semantic vs. memory for morphosyntactic dependencies) might be influenced by sleep in different ways. While this is an interesting possibility, it should be noted that the results regarding sleep characteristics and ERP indices are correlational, leaving open questions with regard to cause and effect (e.g., How can it be ruled out that mnemonically advanced infants simply also show specific sleep features?).

Authors’ response: We agree. For central-parietal sleep spindles, we have already addressed this issue by a within-subject design in one of our studies. In this study, we have shown that an infant’s central-parietal spindle activity is enhanced after lexical-semantic learning compared to comparable stimulation with already known stimuli (i.e., without new lexical-semantic learning), which suggests a causal relationship between sleep spindles and new lexical-semantic memory. Since however, a similar relation has not yet been demonstrated for frontal spindles and specific memory, we can indeed not exclude that individual cognitive or developmental differences contributed to the observed correlation. Therefore, we have included a paragraph on this limitation at page 15:

“At the current stage, however, we cannot rule out the possibility that individual traits or developmental differences between infants have contributed to the relation between left frontal spindles and specific memory. In particular, we do not know whether the observed correlation reflects a direct involvement of sleep spindles in the consolidation process or whether infants with higher spindle activity over left frontal cortex are generally more gifted in consolidating highly specific memories. Despite this limitation, the present data suggests that a mechanism of consolidating specificities in the co-occurrence of items is effectively established at six months of age. Whether this mechanism represents an early form of episodic binding or a kind of implicit associative or holistic memory formation remains to be clarified by future studies.” (15)

(4) From my understanding, this study was not preregistered. Hence, it would be great if the authors could state in more places which of their analyses were confirmatory and which were exploratory in nature.

Authors’ response: We apologize for this missing information in the original manuscript.

Most of our analyses were confirmatory. In particular, we used a similar statistical model (regularity × old/new × laterality × region by group) as in Friedrich et al. 2020 to test expected ERP differences in the memory test of the wake and nap groups. Our correlation analysis with sleep characteristics was also confirmatory, driven by the relationship between spindle activity and N400 memory effect in our

lexical-semantic studies.

However, the subgrouping based on frontal spindle activity (due to the unexpected inverse correlations of the N400-like effect with frontal spindle amplitudes) was exploratory. Now, we have indicated this explorative procedure on page 10: “To further specify the impact of spindle activity on memory, we performed an exploratory analysis by splitting the nap group according to the infants' individual mean left- and mid-frontal spindle amplitudes.” In the method section, we stated on page 26: “For the explorative analyses of the spindle subgroups, we analysed an additional time window with an intermediate latency (600 – 750 ms).”

(5) In the discussion (and perhaps in the introduction), the authors might consider contextualizing their study a bit more. There is a rich developmental literature on early memory that has revealed the high specificity of infant memory (e.g., [https://doi.org/10.1016/0163-6383\(86\)90018-4](https://doi.org/10.1016/0163-6383(86)90018-4) ; <https://doi.org/10.1037/0735-7044.114.1.77> ; [https://doi.org/10.1002/\(SICI\)1098-2302\(199807\)33:1<61::AID-DEV6>3.0.CO;2-Q](https://doi.org/10.1002/(SICI)1098-2302(199807)33:1<61::AID-DEV6>3.0.CO;2-Q). Gradual reduction of specificity of memory retrieval is characteristic of early memory development. While this feature of early memory could be regarded as a deficit as it precludes effective generalization, it also shows that infants do indeed have highly specific memories. This way, this (slightly older) literature would support the authors' notion that formation of specific (or “precise”) memories is a feature of very early cognition.

Authors' response: We thank the reviewer for this note and apologize for the lack of reference in the initial manuscript. We have now included a paragraph in the discussion, in which we refer to the behavioural findings of specific memory in early infancy. We greatly appreciate the studies on infants' earliest memories using deferred imitation and operant mobile/train tasks. However, due to space limitations, we did not go into the details of these studies, but rather discussed a possible involvement of the hippocampus in memory formation at this early age.

“Our finding of specific memory of individual speech phrases in 6- to 8-month-olds joins a large number of behavioural studies showing that early memory is indeed highly specific⁴³⁻⁴⁵. It further extends the existing knowledge in two ways. First, behavioural evidence of specific memory usually comes from the absence of flexible memory use, i.e., a lack of generalization. The present study, however, demonstrates detailed retention of individual speech patterns even despite the extraction of regularities and their generalization to novel verb stems. Second, in the behavioural studies, memory is commonly tested for only one or a very few different events. Here, infants were presented with 16 verb stems, occurring each four times in two different NAD phases in the encoding phase. Even though infants certainly did not retain all of these individual verb phrases, our data strongly suggests that infants with high left frontal spindle amplitudes formed at least some memories with highly overlapping information – an ability, for which the hippocampus is thought to be responsible^{46,47}. This finding challenges the notion that due to the immaturity of the hippocampus, infants in their first year of life are unable to form highly specific memories with overlapping information^{48,49}. Given that functional activity of the immature hippocampus has recently been demonstrated for the encoding of regularities in the temporal occurrence of highly specific visual patterns in infants as young as 3 months of age⁵⁰, it is not entirely impossible that the hippocampus was also involved in the encoding and consolidation of the highly specific overlapping speech patterns in the 6- to 8-month-old infants of the present study.”(p15/16).

Reviewer #3 (Remarks to the Author):

The current manuscript reports the result of one ERP study examine whether infants ages 6- to 8-months retain memory for grammar, and whether post-encoding sleep affects the nature of memory representations. There is a lot to like about the manuscript including the questions beings addressed, the experimental design which is elegant and well controlled, and the focus on neural mechanisms.

Authors' response: We thank the reviewer for her/his appreciation.

(1) The scope of this research, however, is limited by the lack of a behavioral measure of memory retention and generalization (e.g., head-turn paradigm, novelty preference paradigm). The authors correctly state that little is known about whether infants can retain information about grammatical structure and whether sleep modifies the nature of memory representations. Although the reported ERP findings suggest that memory is indeed retained and some generalization occurs, the conclusions are entirely based on the correspondence between the current findings and patterns of event-related potentials gleaned primarily from older groups.

Authors' response: The two main ERP memory effects observed in the wake and nap groups of our study have been repeatedly linked to language and memory in infants of a wide age range, including the age examined in our study. Specifically, the early-latency N200-500 component has been reported in infants from 3 to 20 months of age (e.g., refs. 28-32 of the manuscript). The late memory effect, known as the infant Nc component, has been observed from 3 to at least 12 months of age (refs. 32, 34, 35 – to mention just a few from a large number of studies). Moreover, we related the mid-latency response of our wake group to that of younger (4-months-old) infants (ref. 7) and compared our finding of an N400-like response in the nap group with the N400 of the nap group in a lexical-semantic study at exactly the same age as in the present study (ref. 15). Thus, our conclusions are far from "based on the correspondence between the current findings and patterns of event-related potentials gleaned primarily from older groups," as stated by the reviewer.

(2) Without direct behavioral knowledge, it is entirely possible that these ERP findings do not quite correspond to the same phenomenon in young infants as they do in older infants and adults or do not manifest in overt behaviors that reflect retention. If the latter is the case, then we would have to ask what these pattern of neural activation correspond to and what they afford in terms of memory skill.

Authors' response: While we respect the reviewer's concerns, we would like to emphasize that there is no evidence to date that any of the behavioural methods appropriate for speech stimuli in infants in the studied age range provides a more valid marker of the memory processes of interest than the brain responses measured with ERPs. Head turn/novelty preference procedures in particular rely on involuntary changes in the infants' attention. Given developmental changes in the attentional response (e.g., Houston-Price & Nakai, 2004, *Inf. Child Dev.*, for head turn procedures in general, and Culbertson et al., 2016, *Dev. Psychol.*, for NAD processing in particular), we just do not know at all whether the response measured by these methods reflect the same phenomenon in young infants as they do in older infants, and how it corresponds to overt behaviour of adults.

Nevertheless, (also in response to Reviewer #2's comment) we have now added a paragraph to the discussion that links our ERP results regarding the presence of highly specific memories to previous behavioural studies of infant memory in the first year of (including, in particular, those using deferred imitation tasks for which the measure does not rely on changes in attention)):

“Our finding of specific memory of individual speech phrases in 6- to 8-month-olds joins a large number of behavioural studies showing that early memory is indeed highly specific⁴³⁻⁴⁵. It further extends the existing knowledge in two ways. First, behavioural evidence of specific memory usually comes from the absence of flexible memory use, i.e., a lack of generalization. The present study, however, demonstrates detailed retention of individual speech patterns even despite the extraction of regularities and their generalization to novel verb stems. Second, in the behavioural studies, memory is commonly tested for only one or a very few different events. Here, infants were presented with 16 verb stems, occurring each four times in two different NAD phases in the encoding phase. Even though infants certainly did not retain all of these individual verb phrases, our data strongly suggests that infants with high left frontal spindle amplitudes formed at least some memories with highly overlapping information – an ability, for which the hippocampus is thought to be responsible^{46,47}. This finding challenges the notion that due to the immaturity of the hippocampus, infants in their first year of life are unable to form highly specific memories with overlapping information^{48,49}. Given that functional activity of the immature hippocampus has recently been demonstrated for the encoding of regularities in the temporal occurrence of highly specific visual patterns in infants as young as 3 months of age⁵⁰, it is not entirely impossible that the hippocampus was also involved in the encoding and consolidation of the highly specific overlapping speech patterns in the 6- to 8-month-old infants of the present study.” (p15/16)

(3) I am aware that equating ERP evidence to a cognitive skill, including memory, is not uncommon in the field of infancy. This limitation is typically addressed by exposing infants to experimental conditions that --it is argued--are constrained enough to reveal the predicted skill. However, this approach is not ideal when the fundamental behavioral outcome has not been established in the field (retention of grammar after a delay) and when there is ongoing debate on how to distinguish relevant memory constructs and pinpoint their emergence in early development (i.e., retention of specific details versus retention of general memories; e.g., Kerezstes et al., 2018; Ramsaran et al., 2019). With behavioral data demonstrating a pattern of results consistent with the ERP findings, this manuscript would be more likely to meet the standards of such a high profile journal.

Authors’ response: As the author stated herself/himself in the first paragraph of the review, our study is well controlled. The occurrence of significant ERP differences in the memory test (well-known from a broad range of infant ERP studies) for exactly the same stimuli in same adjacent contexts in the previously learned vs. not learned non-adjacent context conditions must originate from a memory of the nonadjacent relationships, which we believe does not require further support from attention-relying behavioural measures. Our ERP study provides clear evidence for the retention of grammar after a delay, and thus, itself establishes the fundamental outcome in the field.

Our methodological approach even enables us to distinguish between generalised memory for NADs and highly specific memory for individual speech phrases and to demonstrate the existence of both in preverbal infants. In this way, our study makes an important contribution to precisely the ongoing debate about the types of memory in early infancy.

We thank the reviewers for their overall very constructive comments and feel that they have improved our manuscript as a result.

Reviewers' comments:

Reviewer #1 (Remarks to the Author):

I would like to thank the authors for a very thorough and clearly articulated response to the reviews. Most of my comments on the first submission were in relation to clarifying key messages and all of these suggestions have been acted upon satisfactorily. It's also pleasing to see the absence of a correlation between individual retention time and the respective ERP difference for the nap group.

I would also like to add that I agree with the authors that the absence of a behavioural measure alongside the ERP measure does not undermine this study. As stated by the authors, it would be very difficult to incorporate a sensitive behavioural measure at this age, and there's no good evidence to suggest that such a measure would be optimal to ERP measures. In fact, I see the absence of a behavioural measure as a strength, that works to not contaminate the ERP findings.

Finally, the clarification of which analyses were confirmatory and which were exploratory has also greatly improved the transparency of the research.

I stand by my initial position that this study makes a highly valuable theoretical contribution to the literature.

I have no further comments to add.

Reviewer #2 (Remarks to the Author):

The authors have incorporated a number of changes in the revised version of their manuscript. I think that these revisions have further strengthened the manuscript.

In my view, this study makes a significant contribution to our knowledge about the nature of early memory (for language). It is exciting to see more data suggesting that some previous estimates of young infants' mnemonic capabilities (due to immature brain structures) might have been (too) pessimistic.

The suggestion of memory involvement in the nap group is fascinating, although the discussion section around it seems quite speculative in places.

The overall manuscript's contribution to the literature on sleep-dependent memory might be somewhat more limited due to the correlational nature of the findings around spindles/sleep features (and no sleep effect on memory for NADs per se). It is, in my mind, doubtful as to whether the data unequivocally allow conclusions as to which sleep features might be responsible for the potentially observed sleep-dependent memory involvement in the present study. The authors now acknowledge this as a limitation in the discussion section. It is a fundamental issue which will require further research.

I would like to encourage the authors to be careful in their language use in this context, to avoid implying to show causality ("impact") when reporting correlational data ("To further specify the impact of spindle activity on memory,...", p. 10).

Reviewer #3 (Remarks to the Author):

I reviewed the previous version of the manuscript and I continue to think that it is an interesting and well conducted study aimed at using event-related potentials (ERPs) to assess learning and memory of morpho-syntactic, non-adjacent dependencies (NADS) in 6- to 8-month-olds as well as the effects of post-learning nap on retention of such dependencies. Infants were familiarized with NADs, then either napped or stayed awake, and finally they heard sentences that were congruent with the familiarised NADs as well as sentences that violated the familiarised structure. Moreover, half of the test stimuli presented an old intermediate verb stem of the familiarization phase allowing for testing of retention, the other half presented a new intermediate verb stem allowing for testing of generalization. The results show that both retention and generalization occur after either wakefulness or sleep. The indices of retention and generalization are gleaned from ERP data; no behavioral index of the studied phenomena is provided.

In my previous review, I suggested that this lack of behavioral indices is a limitation of the present research. One response from the author is that there is no consensus about an optimal behavioral index. I don't find this argument to be a convincing one because it calls into question the construct being examined. Nevertheless, I am sympathetic to the idea that the examination of the ERP components is interesting in its own right and can provide insight on differences in brain processes induced by experimental conditions. With this in mind, the authors may consider being more careful about using psychological terms to describe their results. It is not appropriate, in my opinion, to describe their effects encoding, retrieval, memory, since we cannot observe these behaviors directly. It would seem more appropriate to describe these as effects of familiarization or re-exposure, which correspond to what was actually done in their experiment.

The manuscript still includes many statements corresponding to interpretations that extend beyond their data. For example, on p. 8 starting from line 183, it is stated that the spatio-temporal similarity of the right-central mid-latency memory effect and the mid-latency effect of familiarity in the wake group "supports the view that infants in the wake group not only retained their memory representations formed in the encoding session, but also used them in the same way when they processed the NADs in the memory test." I do not believe I found any formal assessment of spatio-temporal similarity between these components and there is no assessment of how this information is used by infants. The mere presence of a correlation between two components cannot in itself inform this interpretation.

Another example, on p.9 the paragraph is entitled "Sleep mechanisms of consolidations differ between syntactic and semantic memory", but this study does not examine semantic memory. Although the correlational results reported in the manuscripts show patterns that would indeed suggest differences from semantic memory, the absence of semantic memory condition in this study is consistent with the overreach in interpretation.

Overall, I think that the authors do provide: "clear evidence that 6- to 8-month-old infants retain their familiarity with NADs in memory and generalise the nonadjacent regularities to novel verb stems as indicated by ERP components"; they also show that these effects are robust in that they extend across nap and wake conditions. Other findings are less convincing.

Reviewer #1 (Remarks to the Author):

I would like to thank the authors for a very thorough and clearly articulated response to the reviews. Most of my comments on the first submission were in relation to clarifying key messages and all of these suggestions have been acted upon satisfactorily. It's also pleasing to see the absence of a correlation between individual retention time and the respective ERP difference for the nap group.

I would also like to add that I agree with the authors that the absence of a behavioural measure alongside the ERP measure does not undermine this study. As stated by the authors, it would be very difficult to incorporate a sensitive behavioural measure at this age, and there's no good evidence to suggest that such a measure would be optimal to ERP measures. In fact, I see the absence of a behavioural measure as a strength, that works to not contaminate the ERP findings.

Finally, the clarification of which analyses were confirmatory and which were exploratory has also greatly improved the transparency of the research.

I stand by my initial position that this study makes a highly valuable theoretical contribution to the literature.

I have no further comments to add.

Authors' response: We are very pleased with this evaluation. Many thanks for this appreciation.

Reviewer #2 (Remarks to the Author):

The authors have incorporated a number of changes in the revised version of their manuscript. I think that these revisions have further strengthened the manuscript.

In my view, this study makes a significant contribution to our knowledge about the nature of early memory (for language). It is exciting to see more data suggesting that some previous estimates of young infants' mnemonic capabilities (due to immature brain structures) might have been (too) pessimistic.

Authors' response: We thank the reviewer very much for her/his appreciation.

The suggestion of memory evolution in the nap group is fascinating, although the discussion section around it seems quite speculative in places.

Authors' response: We have carefully reviewed our wording in the relevant paragraphs and have rephrased some sentences (using the subjunctive or modal verbs may and might) to ensure that it is clear when an interpretation around the N400-like memory effect is possible (or even speculative) but not mandatory, e.g., in the result section:

"In the wake group, individual retention time was not correlated with the respective ERP difference ($r = .103$, $P = .542$), rendering unlikely that the overall longer retention time in the nap group was responsible for this effect. The appearance of a new processing stage in the nap group rather suggests that a new form of memory emerged during the sleep period, an effect known as "sleep-dependent memory evolution"³⁶ in adults." (p9 in change tracking view)

and in the discussion: “It might be the case that both the auxiliary/modal verb and the suffixes attained word status, and the order of these words was represented in memory. Alternatively, it could be speculated that the whole AXB and CXD verb phrases were stored as words, in which the middle part (X) was underspecified. ... Regardless of the type of lexicalization that might have occurred in the infants here, the presence of the N400-like effect only in the nap group strengthens the view that sleep promotes the formation of a more mature kind of memory that is usually first observed during later stages in development^{15,37}.” (p14)

“The similarity of the N400-like response with the N400 adults displayed with functioning PFC may even indicate that, although largely immature, the infant PFC already contributes to memory processes in the first year of life. The emergence of a new type of memory in the infants of the nap group might therefore be a first indication for an involvement of the maturing PFC in the sleep-dependent consolidation of infant memory.” (p15)

The overall manuscript’s contribution to the literature on sleep-dependent memory might be somewhat more limited due to the correlational nature of the findings around spindles/sleep features (and no sleep effect on memory for NADs per se). It is, in my mind, doubtful as to whether the data unequivocally allow conclusions as to which sleep features might be responsible for the potentially observed sleep-dependent memory evolution in the present study. The authors now acknowledge this as a limitation in the discussion section. It is a fundamental issue which will require further research.

Authors’ response: We agree with the reviewer and have now added an additional sentence to the respective paragraph: “...In particular, we do not know whether the observed correlations reflect a direct involvement of sleep spindles in the consolidation process or whether infants with higher spindle activity over left frontal and adjacent cortical regions are generally more gifted in consolidating highly specific memories. This fundamental question requires further research and could be explored by within-subject studies that include a baseline assessment of spindle activity, as was done in a previous infant study that focused on the relationship between central-parietal spindles and lexical-semantic memory¹⁶.” (p16)

I would like to encourage the authors to be careful in their language use in this context, to avoid implying to show causality (“impact”) when reporting correlational data (“To further specify the impact of spindle activity on memory,...”, p. 10).

Authors’ response: We carefully checked our wording when referring to correlational data, and in particular have replaced “To further specify the impact of spindle activity on memory,...” by “To further specify the relation between spindle activity and memory,...”. (p11)

Reviewer #3 (Remarks to the Author):

I reviewed the previous version of the manuscript and I continue to think that it is an interesting and well conducted study aimed at using event-related potentials (ERPs) to assess learning and memory of morpho-syntactic, non-adjacent dependencies(NADS) in 6- to 8-month-olds as well as the effects of post-learning nap on retention of such dependencies. Infants were familiarized with NADs, the then either napped or stayed awake, and finally they heard sentences that were congruent with the

familiarised NADs as well as sentences that violated the familiarised structure. Moreover, half of the test stimuli presented an old intermediate verb stem of the familiarization phase allowing for testing of retention, the other half presented a new intermediate verb stem allowing for testing of generalization. The results show that both retention and generalization occur after either wakefulness or sleep. The indices of retention and generalization are gleaned from ERP data; no behavioural index of the studied phenomena is provided.

In my previous review, I suggested that this lack of behavioral indices is a limitation of the present research. One response from the author is that there is no consensus about an optimal behavioral index. I don't find this argument to be a convincing one because it calls into question the construct being examined. Nevertheless, I am sympathetic to the idea that the examination of the ERP components is interesting in its own right and can provide insight on differences in brain processes induced by experimental conditions. With this in mind, the authors may consider being more careful about using psychological terms to describe their results. It is not appropriate, in my opinion, to describe their effects encoding, retrieval, memory, since we cannot observe these behaviors directly. It would seem more appropriate to describe these as effects of familiarization or re-exposure, which correspond to what was actually done in their experiment.

Authors' response: We respect the reviewer's opinion. We did not intend to use the terms encoding, memory, etc. in a strictly behavioural manner, as in our view, these concepts basically refer to the plasticity of the brain and the changes in neural processing triggered by external experiences, and can therefore also be inferred from neural activities. (For example, the replay of neuronal firing patterns in hippocampal place cell assemblies during sleep is commonly taken as evidence of a memory for a specific spatial constellation encoded before sleep).

Nevertheless, while we refer to the first experimental phase as the "encoding phase", we have preferentially used the term "familiarity" when describing the ERP effects of this initial phase. Moreover, we do not use the word "retrieval".

We did not replace the term "memory" because our study was intended to investigate memory and the possible effect of sleep on memory. As we noted in our previous response letter, any significantly different brain response in the memory test phase for exactly the same stimuli in the learned vs. not learned nonadjacent context conditions must originate from a memory of the nonadjacent relations.

The manuscript still includes many statements corresponding to interpretations that extend beyond their data. For example, on p. 8 starting from line 183, it is stated that the spatio-temporal similarity of the right-central mid-latency memory effect and the mid-latency effect of familiarity in the wake group "supports the view that infants in the wake group not only retained their memory representations formed in the encoding session, but also used them in the same way when they processed the NADs in the memory test." I do not believe I found any formal assessment of spatio-temporal similarity between these components and there is no assessment of how this information is used by infants. The mere presence of a correlation between two components cannot in itself inform this interpretation.

Authors' response: We have revised the respective sentence as follows:

"The occurrence of a total of three similar ERP effects in the encoding and memory test phases in the wake group suggests that infants in this group retained their immediately formed representations of the NADs in memory and that the NAD phrases underwent similar processing stages as during the encoding phase when infants were re-exposed to them in the memory test." (p8/9)

Another example, on p.9 the paragraph is entitled “Sleep mechanisms of consolidations differ between syntactic and semantic memory”, but this study does not examine semantic memory. Although the correlational results reported in the manuscripts show patterns that would indeed suggest differences from semantic memory, the absence of semantic memory condition in this study is consistent with the overreach in interpretation.

Authors’ response: In response to the reviewer’s comment, we have changed this subheading to “Relation between nap characteristics and newly evolved memory”. (p10)

In addition, we have removed several interpretative sentences from both the summary of main findings and the results section. We continue to refer to the contrast with previous studies on lexical-semantic learning. However, we have now explicitly stated in the discussion section that our interpretation requires further studies.

“The different effect of sleep on memory for morphosyntactic dependencies in the infants in the present study and on memory for lexical-semantic dependencies in infants of the same age in a previous study¹⁵ give a first indication that the two types of memory do not share the same mechanisms of formation and consolidation. However, studies that directly compare the formation of syntactic and lexical-semantic memory are needed to further support the claim of a difference in these early memory mechanisms.” (p13)

Overall, I think that the authors do provide: “clear evidence that 6- to 8-month-old infants retain their familiarity with NADs in memory and generalise the nonadjacent regularities to novel verb stems as indicated by ERP components”; they also show that these effects are robust in that they extend across nap and wake conditions.

Authors’ response: We are pleased with this consent.

Other findings are less convincing.

Authors’ response: We have now very carefully revised our manuscript and use the subjunctive or modal verbs (may, might, could) to make clear when we merely offer tentative interpretations that do not necessarily follow from the data (see also our responses to the comments of Reviewer #2).

We would like to thank the reviewers once again for their very constructive comments and feel that they have been substantial to improving the manuscript.

Reviewers' comments:

Reviewer #2 (Remarks to the Author):

Thank you to the authors for comprehensively addressing the remaining points and suggestions. In my opinion, the manuscript now provides a balanced report of an interesting and timely study. I have no further comments.

Reviewer #3 (Remarks to the Author):

The authors provided a thorough review and response to comments and I am please to recommend publication.